# CUT LESS, FOLD MORE: MODEL COMPRESSION THROUGH THE LENS OF PROJECTION GEOMETRY

Olga Saukh$^{\circ,\S}$, Dong Wang$^{\circ}$, Haris Šikić$^{\circ}$, Yun Cheng$^{*}$, Lothar Thiele$^{\star}$

$^{\circ}$Graz University of Technology, $^{\S}$Complexity Science Hub, Austria
$^{*}$Swiss Data Science Center, $^{\star}$ETH Zurich, Switzerland

{saukh@, dong.wang@, haris.sikic@student.}tugraz.at,
yun.cheng@sdsc.ethz.ch, thiele@tik.ee.ethz.ch

## ABSTRACT

Compressing neural networks without retraining is vital for deployment at scale. We study calibration-free compression through the lens of projection geometry: structured pruning is an axis-aligned projection, whereas model folding performs a low-rank projection via weight clustering. We formalize both as orthogonal operators and show that, within a rank distance of one, folding provably yields smaller parameter reconstruction error, and under mild smoothness assumptions, smaller functional perturbations than pruning. At scale, we evaluate >1'000 checkpoints spanning ResNet18, PreActResNet18, ViT-B/32, and CLIP ViT-B/32 on CIFAR-10 and ImageNet-1K, covering diverse training hyperparameters (optimizers, learning rates, augmentations, regularization, sharpness-aware training), as well as multiple LLaMA-family 60M and 130M parameter models trained on C4. We show that folding typically achieves higher post-compression accuracy, with the largest gains at moderate–high compression. The gap narrows and occasionally reverses at specific training setups. Our results position folding as a geometry-aware, calibration-free alternative to pruning that is often superior in practice and principled in theory.

## 1 INTRODUCTION

Neural network compression is critical for deploying models in resource-constrained environments. Common approaches include quantization, which reduces the precision of weights and activations, and knowledge distillation, which transfers information from a large teacher model to a smaller student model. In this work, we focus on the class of calibration-free post-training structured compression methods that optimize the model architecture itself without access to training data. Among these, the most widely used is *magnitude-based pruning*, which prunes tensor elements according to their magnitudes, using them as a proxy for their contribution to model accuracy (Han et al., 2015; Mishra et al., 2021; Lu et al., 2023; Ding et al., 2024; Bambhaniya et al., 2024). When combined with fine-tuning or a lightweight BatchNorm reset (Saikumar & Varghese, 2025), this approach achieves significant compression rates with negligible accuracy loss (Kurtic et al., 2022; Sanh et al., 2020). In contrast, the recently introduced *model folding* clusters similar weights and ties them together, providing an approximation of the original network (Wang et al., 2025). Both pruning and folding reduce parameter count but differ fundamentally: pruning removes weights entirely, while folding preserves them in merged representations.

In this work, we develop a unified theoretical and empirical framework to compare pruning and folding through the lens of *orthogonal projections* in parameter space. We show that both compression methods can be viewed as projections onto lower-dimensional subspaces, but with crucial differences in geometry: pruning corresponds to axis-aligned coordinate projections, while folding projects onto cluster-structured subspaces that retain directional information.

At a high level, both pruning and folding compress the weights of a model. We show that for any pruned solution there exists a folded alternative that is *almost* as small—using one extra component

in the compressed representation—yet is strictly closer to the original weights (smaller Frobenius norm), which in turn bounds the change in the network function. Intuitively, folding merges weight vectors with similar directions rather than zeroing coordinates, so the compressed model stays closer in behavior to the initial network.

Empirically, we perform a comprehensive calibration-free study over >1'000 checkpoints spanning CNNs and ViTs on CIFAR-10 and ImageNet-1K, trained under diverse hyperparameter choices (optimizers, learning rates, augmentation, regularization, sharpness-aware training). We also train and process 18 LLaMA-family models with 60M and 130M parameters on C4, by varying learning rates, warmup lengths, and weight decay strength. After compression and also followed by lightweight and full fine-tuning, folding typically attains higher post-compression accuracy, with the largest gains at moderate to high compression. The margin narrows, and can occasionally reverse, at very low compression or under specific training setups, but the overall trend is consistent with our theoretical analysis. Our projection-based perspective opens new directions for designing compression methods that explicitly optimize for functional closeness. This paper makes the following contributions:

- We introduce a unified projection framework that casts pruning and folding as orthogonal projections onto, respectively, axis-aligned and cluster-structured subspaces. We prove that at a compression rank difference of one, folding achieves smaller parameter reconstruction error and tighter function-perturbation bounds under mild smoothness assumptions.
- A large-scale evaluation across >1'000 checkpoints and diverse hyperparameters, covering CNNs and ViTs on CIFAR-10 and ImageNet-1K, as well as LLaMA-60M and LLaMA-130M on C4. In addition, we use post-compression lightweight LayerNorm reset for ViTs, or full-fine-tuning to show that the strong performance of folding is preserved in these settings.
- We show that folding is a geometry-aware alternative that is often superior in practice, with clearly identified regimes (*e.g.*, moderate–high compression) where its advantage is most pronounced, and corner cases where the gap narrows.

We discuss related work in Appendix C, however, the main text already positions pruning and folding within our projection framework and clarifies the novelty of our approach.

## 2 Unified Framework for Pruning and Folding

### 2.1 Preliminaries and Definitions

We consider a neural network with input $x \in \mathbb{R}^d$. We assume ReLU activations and normalization layers (*e.g.*, BatchNorm or LayerNorm) are present.

To develop the theoretical framework, we focus on compressing a single layer at a time. This layer has $p$ inputs and $m$ outputs with its parameters collected in matrix $\mathbf{W} \in \mathbb{R}^{m \times p}$. A row $w(i)$ of $\mathbf{W}$ is denoted as the $i$th parameter vector with individual weights $w(i, j)$. Since all other network parameters are treated as fixed, the network function can be expressed as $f(x; \boldsymbol{W})$, which is trained to minimize a loss function $L(\mathbf{W})$.

We assume that the loss function $L$ is Lipschitz continuous, *i.e.*, there exists a constant $\kappa > 0$ such that

$$|L(\mathbf{W}_1) - L(\mathbf{W}_2)| \leq \kappa \|\mathbf{W}_1 - \mathbf{W}_2\|_F \tag{1}$$

for all admissible parameter matrices $\mathbf{W}_1$ and $\mathbf{W}_2$. The Frobenius norm of a matrix is defined as $\|A\|_F = \sqrt{\sum_{i,j} |a_{ij}|^2}$, that is, the square root of the sum of the squares of its entries, or equivalently, the $\ell_2$-norm of the vectorized matrix. This Lipschitz condition controls the change in loss with respect to parameter perturbations.

**Orthogonal Projection.** We formalize structured pruning and model folding as orthogonal projections in parameter space. A matrix $\mathbf{C} \in \mathbb{R}^{m \times m}$ is an orthogonal projection if $\mathbf{C} = \mathbf{C}^\top = \mathbf{C}^2$, *i.e.*, it is symmetric and idempotent. Such projections map any parameter vector to its closest point (in the Euclidean norm) within a lower-dimensional subspace.

If the columns of $\mathbf{U} \in \mathbb{R}^{m \times k}$ form a basis of a $k$-dimensional subspace, the corresponding orthogonal projection is

$$\mathbf{C} = \mathbf{U}(\mathbf{U}^\top \mathbf{U})^{-1} \mathbf{U}^\top. \tag{2}$$

Equivalently,

$$\mathbf{C}y = \underset{z \in \text{Range}(\mathbf{U})}{\arg\min} \|y - z\|_2$$

meaning $\mathbf{C}y$ is the orthogonal projection of $y$ onto the subspace spanned by $\mathbf{U}$.

## 2.2 Compression as Orthogonal Projection

*Structured pruning.* Pruning can be viewed as a projection onto a coordinate-aligned subspace at the level of neurons, filters, or channels. Assume the layer outputs are ordered so that the last $m - k$ are pruned. The corresponding basis $\mathbf{U}_p$ spans the $k$-dimensional subspace, with projection matrix $\mathbf{C}_p$ and transformed weight matrix $\mathbf{W}_p$:

$$\mathbf{U}_p = \begin{pmatrix} I \\ 0 \end{pmatrix}, \quad \mathbf{C}_p = \begin{pmatrix} I & 0 \\ 0 & 0 \end{pmatrix}, \quad \mathbf{W}_p = \mathbf{C}_p\mathbf{W}. \tag{3}$$

Consequently, the last $m - k$ rows of $\mathbf{W}_p$ are zero, and the corresponding neurons, filters, or channels can be simply removed.

*Model folding.* Folding groups the parameters into $k$ clusters and replaces each cluster with its mean. Depending on the choice of clusters, a different folding results. Folding can be represented as an orthogonal projection onto the $k$-dimensional subspace spanned by $\mathbf{U}_f \in \{0, 1\}^{m \times k}$, where each row contains exactly one nonzero entry indicating the cluster assignment. A cluster $S_j$ comprises all indices of parameter vectors belonging to it; thus, $u_f(i, j) = 1$ if and only if $i \in S_j$.

The projection $\mathbf{C}_f$ defined in Eq. 2 maps each cluster to its mean (Wang et al., 2025). Specifically,

$$\mathbf{W}_f = \mathbf{C}_f\mathbf{W}, \quad \forall i \in S_j : \ w_f(i) = \mu_j, \quad \mu_j = \frac{1}{|S_j|} \sum_{i \in S_j} w(i), \tag{4}$$

where $\mu_j$ is the mean of cluster $j$. After projection, all parameter vectors within a cluster are replaced by their mean, making them identical. As a result, the corresponding layer outputs are also identical, leaving a total of $k$ distinct neurons, filters, or channels. Practically, the identical layer outputs can be joined while adapting the next layer appropriately, see (Wang et al., 2025).

## 2.3 Folding Dominates Pruning

To compare pruning and folding, we first show that for any choice of pruning, there exists a folding that yields a more accurate approximation of the parameter matrix $\mathbf{W}$.

**Theorem 2.1.** *Given any pruning with basis $\mathbf{U}_p$ of rank $0 \leq k_p \leq m-1$ (i.e., at least one parameter vector is pruned), there exists a folding with basis $\mathbf{U}'_f$ and rank $k_f = k_p + 1$ such that*

$$\|\mathbf{W} - \mathbf{W}_p\|_F^2 \geq \|\mathbf{W} - \mathbf{W}'_f\|_F^2,$$

*where $\mathbf{W}_p = \mathbf{C}_p\mathbf{W}$ and $\mathbf{W}'_f = \mathbf{C}'_f\mathbf{W}$, with $\mathbf{C}_p$ and $\mathbf{C}'_f$ denoting the orthogonal projections defined in Eq. 2.*

In the above theorem, $\mathbf{U}'_f$ denotes the constructive clustering obtained by merging all pruned rows into a single additional cluster. The proof is in Appendix B. By the Lipschitz continuity of the loss function in Eq. 1, the superior approximation property of folding implies a tighter bound on the loss difference compared to pruning:

$$|L(\mathbf{W}) - L(\mathbf{W}'_f)| \leq \kappa \|\mathbf{W} - \mathbf{W}'_f\|_F, \quad |L(\mathbf{W}) - L(\mathbf{W}_p)| \leq \kappa \|\mathbf{W} - \mathbf{W}_p\|_F,$$

with

$$\|\mathbf{W} - \mathbf{W}'_f\|_F^2 \leq \|\mathbf{W} - \mathbf{W}_p\|_F^2.$$

Furthermore, the rank difference $k_f = k_p + 1$ between pruning and folding is practically negligible, since in typical scenarios many parameter vectors are pruned. For instance, under a uniform 50% per-layer retention, a ResNet-18 stage with 256 channels keeps $k_p = 128$ (so folding uses $k_f = 129$), and a ViT-B/32 block with width 768 keeps $k_p = 384$ (so $k_f = 385$); the relative increase is just $1/k_p \approx 0.78\%$ and $0.26\%$, respectively—negligible in practice. Moreover, for all layers and architectures we observe that loss and accuracy vary smoothly as the rank increases from $k_p$ to $k_p + 1$,

no jumps in loss or accuracy, and the error difference between ranks $k$ and $k + 1$ is typically much smaller than the difference between pruning and folding at the same rank (see Appendix F.3).

Finally, we show that folding using optimal $k$-means clustering never yields a less accurate approximation of the parameter matrix $\mathbf{W}$ than pruning.

**Theorem 2.2.** *Let $\mathbf{U}_f^\star$ be the basis obtained from an optimal $k$-means clustering with $k_f$ clusters, i.e., the folding clusters are determined by a $k$-means algorithm minimizing the accumulated within-cluster sum of squares. Then, for any pruning with basis $\mathbf{U}_p$ of rank $k_p = k_f - 1$, we have*

$$\|\mathbf{W} - \mathbf{W}_p\|_F^2 \geq \|\mathbf{W} - \mathbf{W}_f^\star\|_F^2,$$

*where $\mathbf{W}_p = \mathbf{C}_p \mathbf{W}$ and $\mathbf{W}_f^\star = \mathbf{C}_f^\star \mathbf{W}$, with $\mathbf{C}_p$ and $\mathbf{C}_f^\star$ denoting the orthogonal projections defined in Eq. 2.*

The proof is given in Appendix B. This result demonstrates that $k$-means folding is not merely a heuristic, but an optimal projection under clustering constraints. Note that the special folding $\mathbf{W}_f'$ in Theorem 2.1 is suboptimal, while Theorem 2.2 shows that $\mathbf{W}_f^\star$ achieves the minimum possible reconstruction error over all clusterings, producing a strictly stronger improvement as $\|\mathbf{W} - \mathbf{W}_p\|_F^2 \geq \|\mathbf{W} - \mathbf{W}_f'\|_F^2 \geq \|\mathbf{W} - \mathbf{W}_f^\star\|_F^2$. Unlike pruning, which relies on parameter vector removal, folding generalizes the idea by enabling coordinated parameter merging. Thus, folding incurs less parameter distortion and provably smaller loss perturbation under a local parameter-Lipschitz assumption.

In addition, Theorem 2.2 has implications for a possible fine-tuning after compression. Matrix $\mathbf{W}$ contains the optimized weights and $\mathbf{W}_p$ or $\mathbf{W}_f^\star$ contain the weights after pruning and folding the optimized network. As a result of Theorem 2.2, the quadratic distance between the optimized weights and the compressed optimized weights is smaller for folding in comparison to pruning.

Our theoretical results employ a one–rank slack comparing pruning at rank $k_p$ to folding at $k_f = k_p + 1$, as a proof device to obtain a clean monotonicity guarantee on projection error. This slack does *not* reflect our evaluation protocol. In all experiments we enforce matched sparsity budgets and compare methods at the *same* retained size (parameters and FLOPs). Hence, empirical accuracy gaps cannot be attributed to extra capacity.

## 3 EXPERIMENTAL RESULTS

Most pruning studies vary only seeds by training several checkpoints under a single hyperparameter recipe, leaving the role of upstream training underexplored. We instead benchmark $> 1'000$ checkpoints spanning diverse hyperparameters (optimizers, learning rates, augmentation, regularization, SAM) to quantify how training choices interact with folding and pruning. Concretely, we train 216 ResNet18 (Adam) and 576 ResNet18 (SGD) models on CIFAR-10, include 50 PreActResNet18 and 200 ViT-B/32 checkpoints from (Andriushchenko et al., 2023), and add 72 CLIP ViT-B/32 models fine-tuned on ImageNet-1K from (Wortsman et al., 2022b). The two ViT families differ markedly in scale ($\sim$19M vs. $\sim$151M parameters). We also train 36 LLaMA-family 60M and 130M parameter models on the Colossal Clean Crawled Corpus (C4) (Raffel et al., 2020). Training details are in Appendix D. The results for LLaMa-130M are in Appendix F.

We empirically compare model folding and structured pruning across CNNs, ViTs and LLaMA-60M models under matched training setups. Unless stated otherwise, we do not apply gradient-based fine-tuning: for CNNs we only re-estimate batch-normalization statistics via a single forward pass using REPAIR (Jordan et al., 2023) to isolate structural effects, and ViTs / LLaMA-60M models are left uncalibrated. Note that REPAIR was recently shown to substantially improve post-compression performance for pruned models (Saikumar & Varghese, 2025), and has also been applied on top of folding (Wang et al., 2025). We report results (i) immediately after compression (CNNs after REPAIR, ViTs with no further step), (ii) for ViTs additionally after a LayerNorm reset, and (iii) for both families after 1–5 epochs of full fine-tuning.

**Folding vs. Structured Pruning on CNNs and ViTs.** We compare model folding (FOLD) with structured magnitude pruning (MAG) under L1 and L2 criteria (MAG1, MAG2) across representative CNN and ViT architectures. Fig. 1 summarizes results: scatter plots show accuracy of MAG1 vs. FOLD for each trained model, with compression ratio indicated by color. Results for MAG2 are in Appendix E.

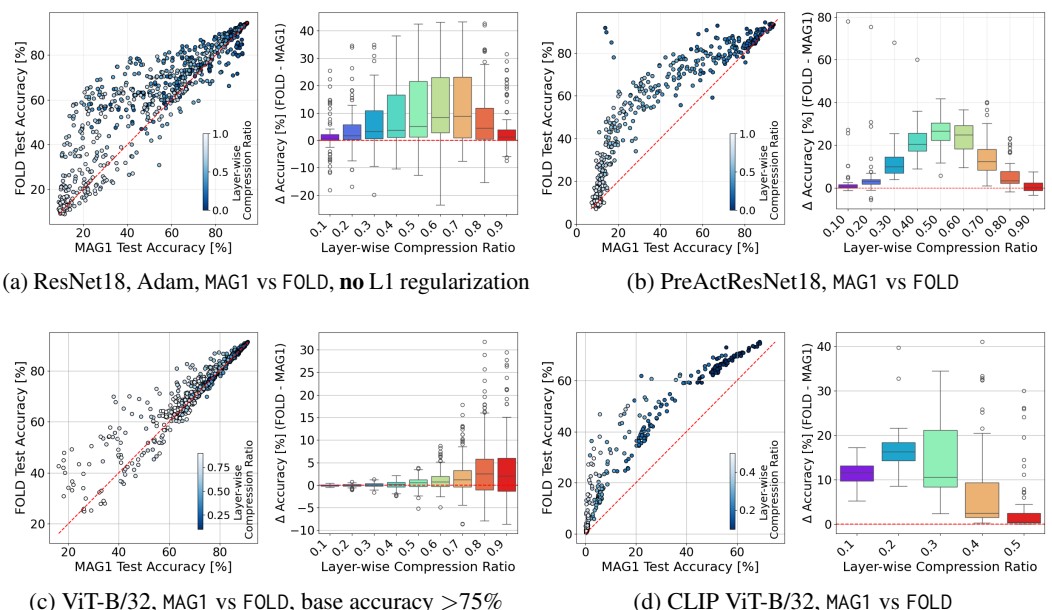

(a) ResNet18, Adam, MAG1 vs FOLD, **no** L1 regularization      (b) PreActResNet18, MAG1 vs FOLD

(c) ViT-B/32, MAG1 vs FOLD, base accuracy >75%      (d) CLIP ViT-B/32, MAG1 vs FOLD

Figure 1: **Folding outperforms magnitude pruning across diverse training regimes. Top row:** ResNet18 and PreActResNet18 on CIFAR-10. ResNet18 checkpoints were trained from scratch with Adam using different hyperparameter configurations. PreActResNet18 checkpoints are from Andriushchenko et al. (2023). **Bottom row:** ViT-B/32 on CIFAR-10 from (Andriushchenko et al., 2023) and CLIP ViT-B/32 on ImageNet-1K from (Wortsman et al., 2022b). See Appendix D for details. In these plots, we use checkpoints that were trained without L1 regularization. Scatter plots show post-compression accuracy for magnitude pruning (L1 criterion) versus folding at uniform per-layer compression ratios. Bar plots depict the accuracy gain by folding, computed as $\Delta = \mathrm{Acc}(\mathtt{FOLD}) - \mathrm{Acc}(\mathtt{MAG1})$, as a function of layer-wise compression ratio. Folding yields the largest improvements at moderate to high compression, confirming its robustness across architectures and datasets. Fig. 8 shows the results for magnitude pruning with L2 criterion.

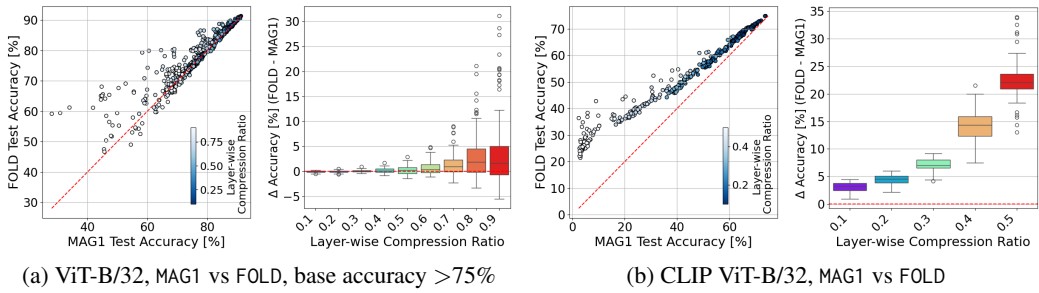

(a) ViT-B/32, MAG1 vs FOLD, base accuracy >75%      (b) CLIP ViT-B/32, MAG1 vs FOLD

Figure 2: **MAG1 versus FOLD on ViTs after LayerNorm-only fine-tuning** for ViT-B/32 on CIFAR-10 and CLIP ViT-B/32 on ImageNet-1K. In the scatter plots, points are checkpoints, color encodes layer-wise compression. Bar plots depict the accuracy gain $\Delta = \mathrm{Acc}(\mathtt{FOLD}) - \mathrm{Acc}(\mathtt{MAG1})$, which is positive, indicating that even under LayerNorm adaptation FOLD retains an advantage over pruning.

Box plots depict the distribution of accuracy differences between FOLD and MAG1. Positive differences indicate folding outperforms pruning, with the gap widening at higher sparsity. This trend holds across ResNet18, PreActResNet18, ViT-B/32, and CLIP ViT-B/32 on both CIFAR-10 and ImageNet-1K, demonstrating robustness to architecture and dataset scale. These results support our theoretical claim (Sec. 2): folding projects onto cluster-structured subspaces, preserving parameter alignment and reducing functional distortion, yielding consistent accuracy gains over magnitude pruning.

**Performance Comparison after Lightweight and Full Fine-Tuning.** The previous results isolate structural effects by evaluating models without further optimization. We now test whether folding's

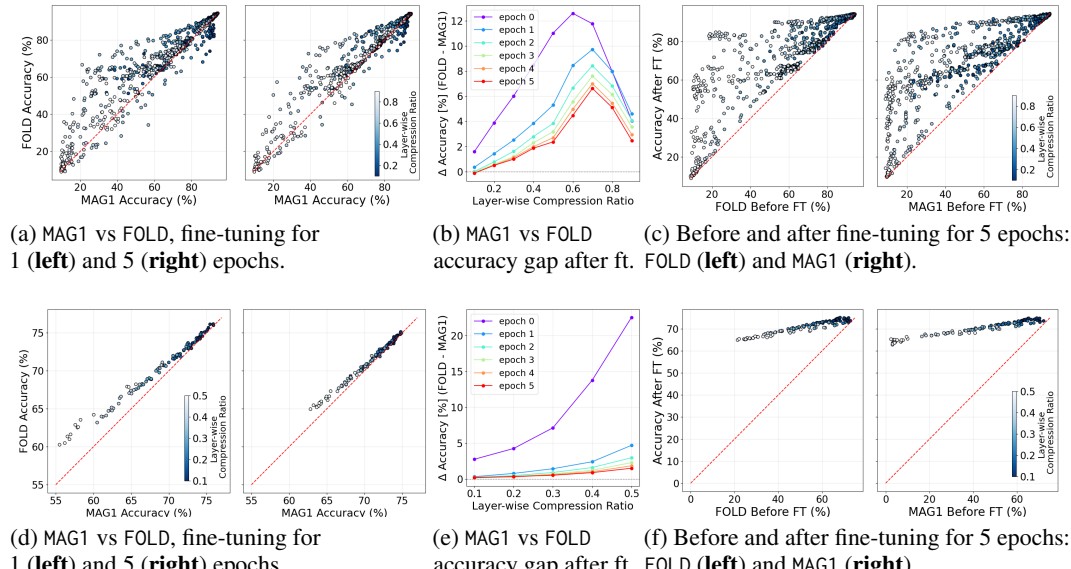

(a) MAG1 vs FOLD, fine-tuning for 1 (**left**) and 5 (**right**) epochs.

(b) MAG1 vs FOLD accuracy gap after ft.

(c) Before and after fine-tuning for 5 epochs: FOLD (**left**) and MAG1 (**right**).

(d) MAG1 vs FOLD, fine-tuning for 1 (**left**) and 5 (**right**) epochs.

(e) MAG1 vs FOLD accuracy gap after ft.

(f) Before and after fine-tuning for 5 epochs: FOLD (**left**) and MAG1 (**right**).

Figure 3: **Folded models retain their accuracy advantage after fine-tuning.** Results for ResNet18 trained by Adam on CIFAR-10 (**top row**) and CLIP-ViT-B/32 trained on ImageNet-1K (**bottom row**): **(a,d)** compares post-compression accuracy of magnitude pruning (MAG1) versus folding (FOLD) after 1 and 5 epochs of fine-tuning. **(b,e)** show the accuracy gap between folding and pruning as a function of fine-tuning epochs, demonstrating that folding maintains a consistent lead, *i.e.*, the FOLD accuracy delta is positive. **(c,f)** illustrate accuracy trajectories before and after 5 epochs of fine-tuning for both methods, highlighting that folded models recover accuracy faster. Further results in Appendix E.

advantage persists after fine-tuning. Fig. 2 compares MAG1 and FOLD on ViTs under lightweight LayerNorm-only adaptation: across ViT-B/32 (CIFAR-10) and CLIP ViT-B/32 (ImageNet-1K), folding consistently reaches higher post-compression accuracy, with the gap $\Delta = \mathrm{Acc}(\texttt{FOLD}) - \mathrm{Acc}(\texttt{MAG1})$ remaining positive and typically growing with compression.

Next, we allow brief fine-tuning (1–5 epochs). Fig. 3 shows that folded models (a,d) start from higher accuracy and retain their lead, (b,e) maintain a positive relative gap, and (c,f) recover faster with

| weight_decay | warmup_steps | max_lr | PPL↓ 0% sparsity | PPL↓ MAG2 (20%) | PPL↓ FOLD (20%) | PPL↓ MAG2 (50%) | PPL↓ FOLD (50%) |
|---|---|---|---|---|---|---|---|
| 0.01 | 880 | 0.001 | 32.11 | 54.51 | **47.17** | 398.62 | **221.32** |
| 0.01 | 1100 | 0.001 | 32.14 | 50.11 | **46.75** | 220.54 | **172.57** |
| 0.01 | 2200 | 0.001 | 32.20 | **46.57** | 47.54 | **174.58** | 216.36 |
| 0 | 880 | 0.001 | 32.17 | 51.14 | **48.23** | 220.33 | 223.86 |
| 0 | 1100 | 0.001 | 32.21 | 50.03 | **47.47** | 231.41 | **204.47** |
| 0 | 2200 | 0.001 | 32.40 | **46.38** | 46.92 | **177.48** | 185.27 |
| 0.01 | 880 | 0.005 | 30.12 | 68.70 | **55.32** | 641.69 | **302.43** |
| 0.01 | 1100 | 0.005 | 29.77 | 68.29 | **49.81** | 564.96 | **234.56** |
| 0.01 | 2200 | 0.005 | 29.60 | 54.50 | **47.04** | 360.52 | **208.02** |
| 0 | 880 | 0.005 | 30.47 | 78.73 | **62.35** | 762.05 | **395.04** |
| 0 | 1100 | 0.005 | 30.17 | 59.20 | **49.58** | 544.87 | **184.74** |
| 0 | 2200 | 0.005 | 29.75 | 56.18 | **46.55** | 353.35 | **165.21** |
| 0.01 | 880 | 0.01 | 31.82 | 66.98 | **51.80** | 910.48 | **406.75** |
| 0.01 | 1100 | 0.01 | 29.85 | 102.41 | **67.69** | 977.92 | **367.94** |
| 0.01 | 2200 | 0.01 | 29.25 | 51.46 | **44.28** | 323.68 | **288.83** |
| 0 | 880 | 0.01 | 108.56 | 129.77 | **123.85** | 279.17 | **198.72** |
| 0 | 1100 | 0.01 | 30.31 | 97.97 | **61.19** | 860.14 | **533.62** |
| 0 | 2200 | 0.01 | 29.57 | 54.43 | **47.77** | 351.11 | **209.06** |

Table 1: **Evaluation of FOLD and MAG2 on LLaMA-60M.** We train and evaluate 18 LLaMA-family models with 60M parameters on C4 by varying max_lr, warmup steps and weight decay. Columns 4–8 show perplexity of the trained model before compression and after pruning / folding using layer-wise pruning ratio of $20\%$ and $50\%$. We prune only FFN blocks. Except for low learning rates with long warmup schedules, FOLD outperforms MAG2 (highlighted in bold).

fewer plateaus. Thus, folding provides a better initialization and requires fewer updates to regain performance, making it advantageous in settings with limited fine-tuning.

**Performance Comparison on LLaMA-60M.** Tab. 1 evaluates `FOLD` and `MAG2` on LLaMA-60M trained on C4 under 18 hyperparameter settings (varying learning rate, warmup, and weight decay). We prune or fold only the FFN blocks and report perplexity at baseline and at 20% and 50% layer-wise sparsity. Except for models trained with very low learning rates and long warmup, `FOLD` consistently outperforms `MAG2`. Similar finding have been obtained for LLaMA-130M models in Tab. 6.

## 4 MODEL COMPRESSION ABLATION STUDIES

The previous sections demonstrated that folding often outperforms structured pruning across architectures and compression ratios. On ResNets and ViTs, we probe which training factors impact this advantage. Specifically, we analyze sensitivity to learning rate, the use of sharpness-aware training (SAM) (Foret et al., 2021), regularization and data augmentation (Prabhu et al., 2019)—the hyperparameters known to influence loss landscape geometry and generalization (Fort & Jastrzebski, 2019; Li et al., 2018; Neyshabur et al., 2017; Chen et al., 2022) in non-trivial ways (Andriushchenko et al., 2023). To validate these curvature-related hypotheses, Appendix F includes a sharpness analysis. Our measurements quantify how hyperparameters shift the local geometry of the loss landscape and help explain when `FOLD` 's advantage widens or narrows.

**Role of Optimizer.** We repeat the ResNet18 analysis under Adam and SGD to gauge optimizer sensitivity. Compared to the Adam-trained sweep in Fig. 1(a), the complementary SGD sweep in Fig. 5 shows the same qualitative ordering—`FOLD` exceeds `MAG1` across compression levels—but with different baselines and dispersion: SGD checkpoints form a tighter cloud and exhibit a smaller median gap, whereas Adam yields larger variance and at times a more pronounced `FOLD` advantage, especially at higher compression. The `FOLD`–`MAG1` difference remains positive under both optimizers in most cases, but its magnitude is optimizer-dependent.

**Effect of Learning Rate.** Fig. 4 reports post-compression accuracy for `FOLD` versus `MAG1` across learning rates on ResNet18 (Adam, SGD), PreActResNet18, and ViT-B/32. With Adam, `FOLD` 's edge is largest at moderate–low rates, narrows and can reverse at very high rates, and vanishes again at extremely small rates (both methods degrade). For SGD, the dependence is weaker and can be inverted (*e.g.*, ViT-B/32). The effect of learning rate is expressed through sharpness (see Appendix F): when training places the model in regions where folding produces a smaller sharpness increase than pruning, folding wins. When folding produces a larger sharpness increase (most visible at high learning rates under Adam), pruning can outperform. Adaptive methods like Adam are associated with sharper minima and distinct generalization behavior compared to SGD, amplifying this sensitivity (Wilson et al., 2018; Jastrzębski et al., 2018; Zhou et al., 2021).

**Effect of SAM.** Fig. 6 evaluates training with and without SAM and measures post-compression accuracy. Across models, SAM lifts both methods. With light L1 regularization ($10^{-5}$) during training shown in (b), pruning narrows the gap at *low* compression (where induced sparsity aligns with L1), yet `FOLD` regains and extends its lead as compression increases. These trends are consistent with the view that SAM steers training to flatter solutions, reducing curvature sensitivity. Within this flatter neighborhood both pruning and folding projections operate inside the same robustness ball, so their

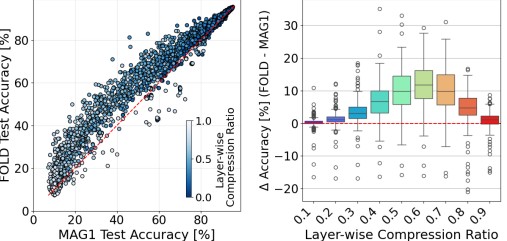

Figure 5: **Optimizer effect** evaluated on ResNet18 checkpoints trained on CIFAR-10 with SGD (no L1 regularization). The figure complements Fig. 1(a).

geometric differences matter less and the gap narrows—an effect stronger for ViT-B/32, where high $\rho$ homogenizes head / channel saliencies and reduces the relative advantage of clustering.

**Effect of Data Augmentation.** Fig. 7 plots the distribution of $\Delta$Accuracy (`FOLD` − `MAG1`) across checkpoints versus the layer-wise compression ratio, contrasting runs without (gray) and with RandAugment (green). For ResNet18 (Adam and SGD) and PreActResNet18, RAUG reduces or shifts `FOLD` 's relative benefit. In contrast, for ViT-B/32 RAUG increases `FOLD` 's advantage: the median $\Delta$

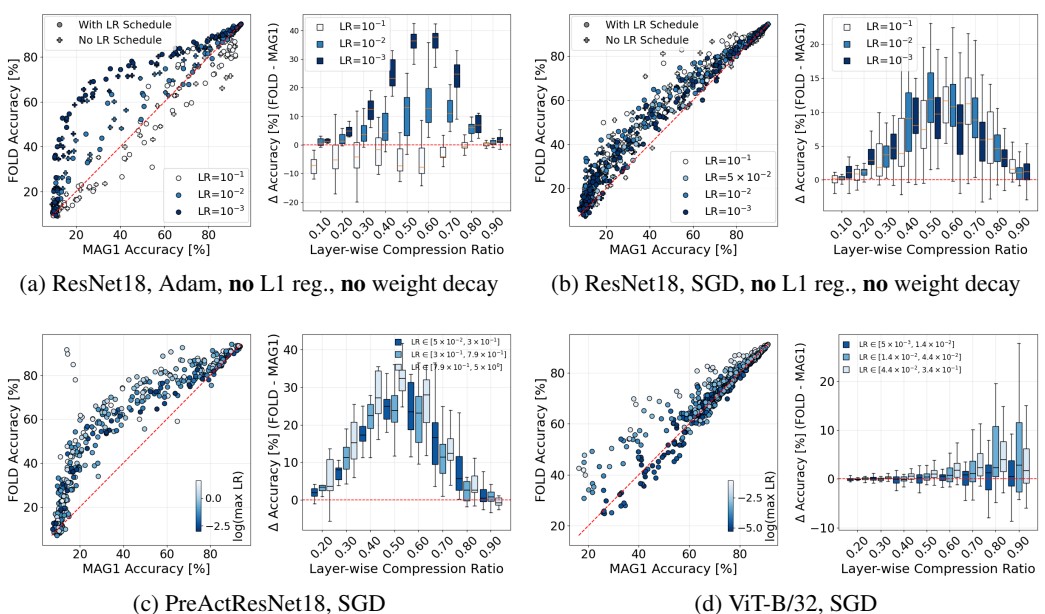

(a) ResNet18, Adam, **no** L1 reg., **no** weight decay

(b) ResNet18, SGD, **no** L1 reg., **no** weight decay

(c) PreActResNet18, SGD

(d) ViT-B/32, SGD

Figure 4: **Learning rate modulates folding's edge.** Post-compression accuracy of FOLD and MAG1 across learning rates: ResNet18 with Adam **(a)** and SGD **(b)**, PreActResNet18 **(c)**, and ViT-B/32 **(d)**. FOLD leads at moderate–low rates. With Adam, the gap shrinks or reverses at very high rates, and closes again at extremely small rates. SGD shows weaker or opposite dependence.

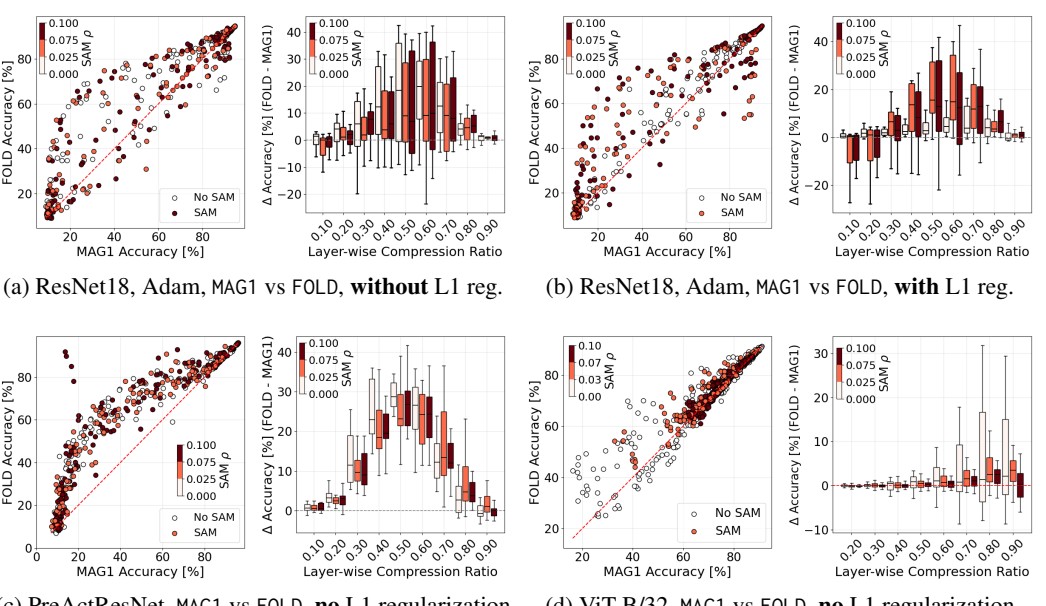

(a) ResNet18, Adam, MAG1 vs FOLD, **without** L1 reg.

(b) ResNet18, Adam, MAG1 vs FOLD, **with** L1 reg.

(c) PreActResNet, MAG1 vs FOLD, **no** L1 regularization

(d) ViT-B/32, MAG1 vs FOLD, **no** L1 regularization

Figure 6: **SAM (Foret et al., 2021) can boost model compression.** Post-compression accuracy under training with / without SAM. **(a)** ResNet18 (Adam), no L1. **(b)** ResNet18 (Adam), L1= $10^{-5}$. **(c)** PreActResNet18 (SGD), no L1. **(d)** ViT-B/32, no L1. SAM improves both FOLD and MAG1, but the uplift is consistently larger for FOLD, especially with Adam. Light L1 regularization helps MAG1 at low compression, yet FOLD retains a clear advantage at moderate–high compression.

rises with compression, suggesting that augmented ViT representations are especially amenable to projection-based removal. A plausible mechanism is that augmentation biases training toward flatter, more invariant solutions. This is supported by our sharpness analysis in Appendix F and consistent

with recent theory linking augmentation-induced input perturbations to equivalent parameter-space perturbations and showing that augmentations bias training toward flatter minima (Yoo & Yoon, 2025). Standard augmentation (`augm=True`) shows a similar trend and is omitted for brevity.

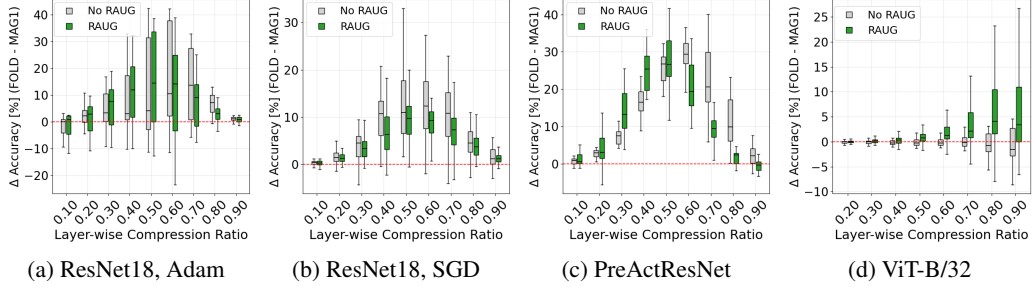

(a) ResNet18, Adam    (b) ResNet18, SGD    (c) PreActResNet    (d) ViT-B/32

Figure 7: **Augmentations have a generally positive effect on the post-compression accuracy.** Post-compression accuracy without / with random augmentations for **(a)** ResNet18 (Adam), **(b)** ResNet18 (SGD), **(c)** PreActResNet18, and **(d)** ViT-B/32. Augmentations boost both `FOLD` and `MAG1`. On ResNet18 they also narrow `FOLD`'s advantage consistent with added invariances making axis-aligned removals less damaging. In ViT-B/32, augmentations are essential for folding[1].

These ablations reveal a consistent pattern: conditions that encourage flatter and structured solutions—moderately low learning rates and SAM with a small–moderate radius—magnify `FOLD`'s advantage, whereas extremes reduce it: very high or very low learning rates, stronger augmentations, or large SAM radii narrow the gap; SGD generally dampens all effects relative to Adam. This aligns with our projection view (Sec. 2): when weights are well aligned, clustering reduces projection error more than coordinate removal and thus perturbs the function less, while weaker alignment or broad robustness neighborhoods make the two projections behave more similarly.

## 5    CONCLUSION, LIMITATIONS, AND OUTLOOK

We framed structured pruning and model folding as projection-based compression and showed that folding achieves smaller parameter deviation with a one-rank slack, implying tighter functional preservation under mild smoothness. A calibration-free evaluation over $>1'000$ checkpoints (ResNet18, PreActResNet18, ViT-B/32, CLIP ViT-B/32; CIFAR-10, ImageNet-1K; and LLaMA-60M and LLaMA-130M on C4) found that `FOLD` typically surpasses `MAG1` in post-compression accuracy, with the clearest gains at moderate–high compression and under training conditions that induce flatter, more structured solutions (*e.g.*, moderate learning rates, SAM). The gap narrows at very low compression and can shrink under strong data augmentation or large SAM radii.

**Limitations.** Our theoretical guarantee allows a one-component increase in compressed rank but does not establish universal dominance at exactly matched sizes. Empirically, we focus on standard CNN and ViT families on CIFAR-10 and ImageNet-1K, as well as small LLaMA models on C4. For ViTs and LLaMA, pruning and folding are applied only to the FFN blocks. Extensions to attention layers is left for future work. We evaluate in strictly calibration-free settings, with optional BatchNorm/LayerNorm resets and short fine-tuning budgets, and compare primarily against magnitude-based structured pruning. Interactions with quantization, distillation, and unstructured sparsity are not considered. Larger LLMs are beyond the scope of this study due to the computational cost of training across diverse hyperparameter settings. We note that most SoTA pruning methods for LLMs rely on calibration data (*e.g.*, activation-aware/second-order) and are exclusively pruning-based.

**Outlook.** We plan to extend folding to pruning / folding attention blocks, calibration-based settings and evaluate on larger LLMs / VLMs. We also plan to study interactions with quantization and adaptation methods. More broadly, our projection-based view positions folding as a geometry-aware primitive for compression: a foundation on which novel calibration-based compression methods, hybrid pipelines with quantization and distillation can be built, and a step toward principled model compression framework. In this sense, folding is not only a practical tool but a building block for the next generation of compression methods tailored to foundation models and deployment at scale.

---

[1]Note that the base accuracy of ViT-B/32 checkpoints trained without RAUG is lower than with RAUG.

**Reproducibility Statement.** Our compression operators and evaluation protocol are described in Sec. 2-Sec. 3, with ablation studies in Sec. 4. A repository with configs and scripts to regenerate all figures/tables is linked in Appendix A. Complete proofs are in Appendix B. Related literature is summarized in Appendix C. Training setups, datasets, links to the used checkpoints, and hyperparameter grids in Appendix D, and extended results in Appendix E and Appendix F. Our limited LLM usage statement is in Appendix G. Together, these materials enable re-running the full pipeline and regenerating the results.

**Acknowledgments.** This work was in part supported by the FFG COMET K1 Centre "Pro$^2$Future II" (Cognitive and Sustainable Products and Production Systems of the Future; Contract No. 911655). The results presented in this paper were obtained using the computational resources of Pro2Future GmbH, and the Austrian Scientific Computing (ASC) infrastructure.

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

APPENDIX

The following sections provide supplementary information and complement the main paper:

## A    CODE, DATA, AND RESOURCES

**Code and logs.**  A repository with all source code, experiment configs, and figure-generation scripts (including the exact logs used to render every plot/table) are released at `https://github.com/osaukh/folding_as_projection`. The repo contains: implementations of folding and pruning operators, training/evaluation pipelines, scripts to plot ablations, and notebooks to reproduce figures directly from logs. We log all training metrics and hyperparameters with Weights & Biases[2] and export logs alongside the code for reproduction. Additionally, we provide another repository for reproducing results of compressing LLaMA-60M and LLaMA-130M with folding and magnitude structured pruning at `https://github.com/nanguoyu/simple_model_folding_public`.

Our folding implementation is based on the code by Wang et al. (2025)[3].

**Datasets.**  We use CIFAR-10[4] and ImageNet-1K[5]. CIFAR-10 is downloaded automatically via `torchvision`. ImageNet-1K requires the official credentials and follows its license. Pretrained/fine-tuned checkpoints referenced in the paper are either trained by us (configs in the repo) or obtained from the cited works (Andriushchenko et al., 2023; Wortsman et al., 2022b). The download links are also provided in Appendix D.

**Compute resources.** Experiments were run on a cluster featuring $8\times$ NVIDIA A100 (80 GB RAM) GPUs. All random seeds are fixed in the configs and scripts.

**Computational complexity and memory cost.** At inference and matched retained sizes, folding and structured pruning yield the same compute and memory. The difference lies in the compression step: magnitude pruning is a one-pass scoring and selection procedure ($O(pm)$ to score $p$ filters of dimension $m$, plus $O(p\log p)$ selection), whereas folding runs $k$-means on layer weights with $T$ sweeps. Using Hartigan's algorithm (Hartigan & Wong, 1979), one sweep costs $O(pkm)$, with max $T = 10$ sweeps the total is $O(pkmT)$ (effectively linear in $pm$ when $k$ is small). This cost is paid once per layer and is small compared to training.

**Runtime overview.** The most expensive step in our study is fine-tuning of CLIP VIT-B/32 on ImageNet-1K (1–5 epochs), which dominates wall-clock time (order of hours per run). In contrast, compression is lightweight. We detail measured runtime overhead of compression in Appendix F.

## B    PROOFS OF THEORETICAL CLAIMS

Below we prove that for any choice of pruning, there exists a folding that yields a more accurate approximation of the parameter matrix $\mathbf{W}$.

---

[2]Weights & Biases: `https://wandb.ai`

[3]Model folding universal: `https://github.com/nanguoyu/model-folding-universal` and model folding for CNNs: `https://github.com/marza96/ModelFolding/`

[4]CIFAR-10: `https://www.cs.toronto.edu/~kriz/cifar.html`

[5]ImageNet-1K: `https://image-net.org/`

**Theorem 2.1.** *Given any pruning with basis $\mathbf{U}_p$ of rank $0 \leq k_p \leq m-1$ (i.e., at least one parameter vector is pruned), there exists a folding with basis $\mathbf{U}'_f$ and rank $k_f = k_p + 1$ such that*

$$\|\mathbf{W} - \mathbf{W}_p\|_F^2 \geq \|\mathbf{W} - \mathbf{W}'_f\|_F^2,$$

*where $\mathbf{W}_p = \mathbf{C}_p \mathbf{W}$ and $\mathbf{W}'_f = \mathbf{C}'_f \mathbf{W}$, with $\mathbf{C}_p$ and $\mathbf{C}'_f$ denoting the orthogonal projections defined in Eq. 2.*

*Proof.* The rows of $\mathbf{W}$ can be ordered such that the pruned parameter vectors are first: $w(1), ..., w(m - k_p)$. Then we find that

$$\mathbf{W} - \mathbf{W}_p = \begin{pmatrix} w(1) \\ \cdots \\ w(m-k_p) \\ 0 \\ \cdots \\ 0 \end{pmatrix}$$

using Eq. 3. For the existence proof, we choose a folding that clusters all parameter vectors $w(1), ..., w(m - k_p)$ into a single cluster, all other parameter vectors have individual clusters, *i.e.,*

$$\mathbf{U}'_f = \begin{pmatrix} 1 & 0 \\ \cdots & 0 \\ 1 & 0 \\ 0 & \mathbf{I} \end{pmatrix} \quad ; \quad \mathbf{W} - \mathbf{W}'_f = \begin{pmatrix} w(1) - \mu \\ \cdots \\ w(m-k_p) - \mu \\ 0 \\ \cdots \\ 0 \end{pmatrix} \quad ; \quad \mu = \frac{1}{m-k_p} \sum_{i=1}^{m-k_p} w(i)$$

using Eq. 4.

We have $\|\mathbf{W} - \mathbf{W}_p\|_F^2 = \sum_{i=1}^{m-k_p} w(i)^T w(i)$ and

$$\|\mathbf{W} - \mathbf{W}'_f\|_F^2 = \sum_{i=1}^{m-k_p} (w(i) - \mu)^T (w(i) - \mu) = \sum_{i=1}^{m-k_p} \left( w(i)^T w(i) - 2 w(i)^T \mu + \mu^T \mu \right)$$

$$= \sum_{i=1}^{m-k_p} w(i)^T w(i) - (m - k_p) \mu^T \mu$$

$$\leq \sum_{i=1}^{m-k_p} w(i)^T w(i) = \|\mathbf{W} - \mathbf{W}_p\|_F^2$$

The latter inequality directly establishes the theorem. □

The following theorem shows that folding using optimal $k$-means clustering never yields a less accurate approximation of the parameter matrix $\mathbf{W}$ than pruning.

**Theorem 2.2.** *Let $\mathbf{U}_f^\star$ be the basis obtained from an optimal $k$-means clustering with $k_f$ clusters, i.e., the folding clusters are determined by a $k$-means algorithm minimizing the accumulated within-cluster sum of squares. Then, for any pruning with basis $\mathbf{U}_p$ of rank $k_p = k_f - 1$, we have*

$$\|\mathbf{W} - \mathbf{W}_p\|_F^2 \geq \|\mathbf{W} - \mathbf{W}_f^\star\|_F^2,$$

*where $\mathbf{W}_p = \mathbf{C}_p \mathbf{W}$ and $\mathbf{W}_f^\star = \mathbf{C}_f^\star \mathbf{W}$, with $\mathbf{C}_p$ and $\mathbf{C}_f^\star$ denoting the orthogonal projections defined in Eq. 2.*

*Proof.* According to Bauckhage (2015) and Wang et al. (2025), the problem of $k$-means clustering can be formulated as the following constrained matrix factorization problem:

$$\min_{\mathbf{U}} \left\| \mathbf{W} - \mathbf{U}(\mathbf{U}^\top \mathbf{U})^{-1} \mathbf{U}^\top \mathbf{W} \right\|_F^2 \quad \text{subject to} \quad u(i,j) \in \{0,1\}, \ \sum_j u(i,j) = 1 \ \forall i.$$

This formulation coincides with the orthogonal projection of model folding, see Eq. 2 and Eq. 4. Theorem 2.1 guarantees the existence of a folding basis $\mathbf{U}'_f$ and the corresponding projection $\mathbf{C}'_f$ for any pruning $\mathbf{W}_p$ of $\mathbf{W}$, such that

$$\|\mathbf{W} - \mathbf{W}_p\|_F^2 \quad \geq \quad \|\mathbf{W} - \mathbf{W}'_f\|_F^2.$$

Since optimal $k$-means clustering achieves the minimal possible error $\|\mathbf{W} - \mathbf{W}_f^\star\|_F^2 \quad \leq \quad \|\mathbf{W} - \mathbf{W}'_f\|_F^2$, the theorem follows. $\qquad\square$

## C  RELATED WORK

Model compression encompasses a wide range of approaches designed to reduce inference cost while preserving model utility. We focus on *post-training, calibration-free structured compression*, where the model architecture is modified without access to data or gradients. In this setting, the dominant baselines are structured pruning and, more recently, model folding. Below we discuss these families of methods and clarify how our projection-theoretic view relates to and extends prior work.

**Post-training compression.** Quantization reduces arithmetic precision (Darvish Rouhani et al., 2020; Qian Zhang et al., 2022), but typically requires calibration to maintain activation ranges. Knowledge distillation (Hinton et al., 2015) produces reduced students trained to imitate teacher logits. Even data-free variants (Micaelli & Storkey, 2019; Chen et al., 2019; Fang et al., 2020; Yu et al., 2023; Haroush et al., 2020) require full training dynamics and do not yield structural compression. Low-rank factorization via matrix or tensor decompositions (Ren & Zhu, 2023; Horvath et al., 2024; Lebedev et al., 2015; Kim et al., 2016) approximates pretrained weights by continuous subspaces but generally requires fine-tuning for restoration. These approaches differ fundamentally from our objective: they modify numerical precision or parameterization, not the discrete structure of the model.

**Structured pruning.** Structured pruning removes neurons, channels, filters, or blocks (Li et al., 2016; Luo et al., 2017; Hu et al., 2016; Wen et al., 2016). Magnitude-based criteria (Han et al., 2015; Lu et al., 2023; Ding et al., 2024; Entezari & Saukh, 2020) dominate due to simplicity and hardware alignment. However, structured pruning typically requires fine-tuning or recalibration (Kurtic et al., 2022; Sanh et al., 2020) to mitigate accuracy degradation, and even calibration-based methods such as SparseGPT (Frantar & Alistarh, 2023) or Wanda (Sun et al., 2024) operate through axis-aligned removal of coordinates. One-shot improvements using N:M sparsity (Yao et al., 2019; Kang, 2020) or OT-based structural alignment (Theus et al., 2024) still operate within the same paradigm: pruning corresponds to enforcing that the retained parameter vectors lie in a fixed coordinate-aligned subspace.

Our work shows that such axis-aligned projections are geometrically restrictive. We formalize pruning as an orthogonal projection onto a coordinate subspace and demonstrate that, at matched ranks up to one slack, pruning is provably dominated by projections onto cluster-structured subspaces.

**Weight clustering and model folding.** Model folding, recently introduced by Wang et al. (2025), ties groups of similar channels by replacing them with their mean, yielding dense low-rank layers that preserve structural couplings. Folding implicitly performs a *cluster-structured projection* determined by discrete assignments, and practical implementations rely on k-means clustering. This operator class is strictly richer than axis-aligned pruning: folding enables coordinated merging rather than coordinate removal, while remaining compatible with dense inference. IFM (Chen et al., 2023) is related in that it also merges channels via grouping, but its variance-collapse correction is ineffective (Wang et al., 2025), leading to substantially weaker performance.

Our work strengthens this line along two axes. First, we provide a unified *projection-geometric framework* showing that both pruning and folding are orthogonal projections, but onto fundamentally different subspaces: coordinate-aligned versus cluster-structured. Second, we prove that for any pruned solution of rank $k$, there exists a folded solution of rank $k+1$ with strictly smaller parameter reconstruction error, and that optimal k-means folding minimizes this projection error among all cluster-structured projections. This establishes a strict theoretical separation between pruning and folding and explains the empirical superiority of folding in calibration-free settings.

**Model merging and alignment.** Model merging combines independently trained models via parameter averaging or permutation alignment. Model soups (Wortsman et al., 2022a) exploit shared initialization. Permutation matching (Entezari et al., 2022; Ainsworth et al., 2023) constructs neuron

correspondences. REPAIR (Jordan et al., 2023) stabilizes fused models by re-normalizing preactivations. Intra-model merging approaches such as ZipIt! (Stoica et al., 2024) combine computational units but do not target compression under fixed architectural constraints.

These works differ from ours in both objective and mechanism. Merging seeks functional fusion across networks, whereas folding compresses a *single* network by exploiting intra-layer redundancy. Our projection-theoretic formulation shows that folding operates as a structured projection with explicit geometric optimality guarantees—properties not shared by merging methods.

**Positioning of this work.** Across pruning, folding, and merging, prior efforts lack a unifying mathematical framework that characterizes the geometry of post-training structural compression. Our contribution is to introduce such a framework: we cast pruning and folding as orthogonal projections and show that cluster-structured projections admit strictly smaller distortion than coordinate projections under practically negligible rank slack. This perspective yields nontrivial theoretical guarantees and aligns closely with the empirical phenomena observed across CNNs, ViTs, and LLaMA models.

## D  TRAINING DETAILS

The following subsections detail the hyperparameters used to train our checkpoints. For checkpoints taken from the literature, we summarize the available training details.

### D.1  RESNET18 ON CIFAR-10 TRAINING SETUP WITH ADAM AND SGD

We trained a total of 792 ResNet18 models on CIFAR-10 by varying hyperparameter configurations. We used two optimizers: Adam and SGD. Tab. 2 summarizes the parameter combinations explored for each optimizer. For Adam, we used 3 learning rates and 1 momentum value. For SGD, we used 3 learning rates and 2 momentum values. The remaining parameters were shared across both optimizers: weight decay (3 values), L1 regularization (2 values), RandAugment (2 values), Sharpness-Aware Minimization (3 values), and learning rate scheduling (2 values). This resulted in 216 models trained with Adam and 576 models trained with SGD. In the ablation studies, we filter checkpoints (as specified in the figure captions) to highlight the observed effects.

| Parameter | Values |
|---|---|
| Optimizer | adam, sgd |
| Learning Rate | adam: 0.1, 0.01, 0.001 |
|  | sgd: 0.1, 0.05, 0.01, 0.001 |
| Momentum | adam: 0.0 |
|  | sgd: 0.9, 0.99 |
| Weight Decay | 0.0, 0.0005, 0.001 |
| L1 Regularization | $0.0, 1 \times 10^{-5}$ |
| RandAugment | True, False |
| SAM (Sharpness-Aware Minimization) | None, 0.05, 0.1 |
| Learning Rate Schedule | True, False |

Table 2: Hyperparameter combinations used for ResNet18 training on CIFAR-10.

### D.2  PREACTRESNET18 ON CIFAR-10

We use 50 trained PreActResNet18 models on CIFAR-10 from (Andriushchenko et al., 2023)[6]. The models are trained using a fixed set of training parameters and a sweep over a few key hyperparameters. Tab. 3 summarizes varied parameters used in this experiment. All checkpoints used the same training protocol: 200 epochs, batch size 128, and no label noise. The model width was fixed at 64 and the learning rate schedule followed a cyclic pattern. Only the maximum learning rate (`lr_max`), SAM strength (`sam_rho`), and augmentation settings were varied. For the learning rate ablation studies, we adopt the reported maximum learning rate.

---

[6]Download link: `https://drive.google.com/drive/folders/1LmthJCb3RXBFWjeTOC4UOOl7Ppgg2h7n`

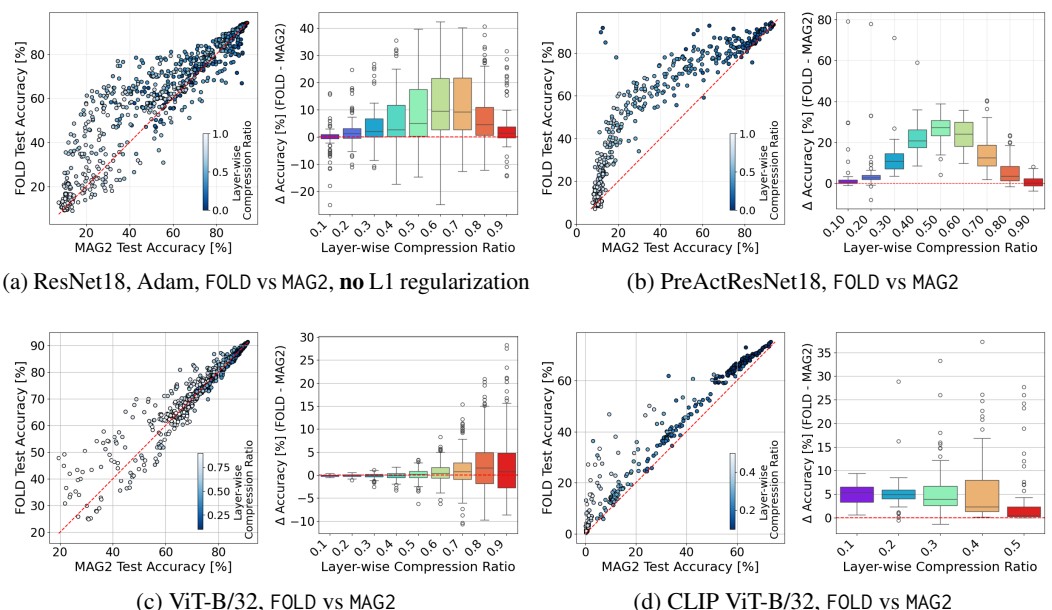

(a) ResNet18, Adam, FOLD vs MAG2, **no** L1 regularization

(b) PreActResNet18, FOLD vs MAG2

(c) ViT-B/32, FOLD vs MAG2

(d) CLIP ViT-B/32, FOLD vs MAG2

Figure 8: **Folding outperforms magnitude pruning across diverse training regimes.** The same setup as in Fig. 1, but compared to the L2 magnitude pruning criterion. **Top row:** ResNet18 and PreActResNet18 on CIFAR-10. ResNet18 checkpoints were trained from scratch with Adam using different hyperparameter configurations. **Bottom row:** ViT-B/32 on CIFAR-10 and CLIP ViT-B/32 on ImageNet-1K. Scatter plots show post-compression accuracy for folding versus magnitude pruning (L2 criterion) at uniform per-layer compression ratios. Bar plots depict the accuracy gain by folding, computed as $\Delta = \mathrm{Acc}(\mathtt{FOLD}) - \mathrm{Acc}(\mathtt{MAG2})$, as a function of layer-wise compression ratio. Folding yields the largest improvements at moderate to high compression, confirming its robustness across architectures and datasets.

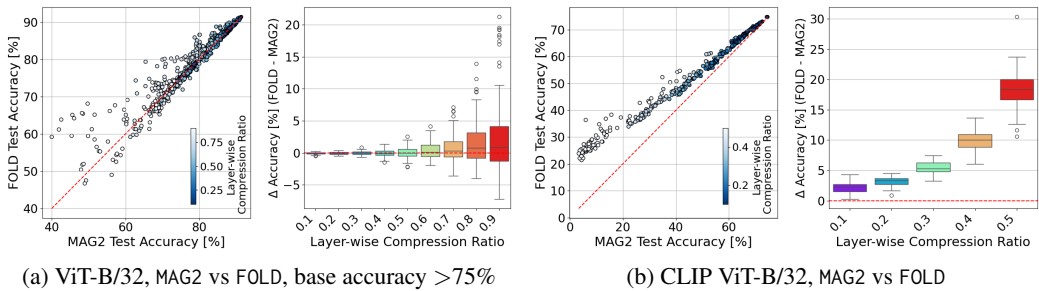

(a) ViT-B/32, MAG2 vs FOLD, base accuracy >75%

(b) CLIP ViT-B/32, MAG2 vs FOLD

Figure 9: **FOLD versus MAG2 on ViTs after LayerNorm-only fine-tuning** for ViT-B/32 on CIFAR-10 and CLIP ViT-B/32 on ImageNet-1K. In the scatter plots, points are checkpoints, color encodes layer-wise compression. Bar plots depict the accuracy gain $\Delta = \mathrm{Acc}(\mathtt{FOLD}) - \mathrm{Acc}(\mathtt{MAG1})$, which remains positive and typically grows with compression, indicating that even under lightweight LayerNorm adaptation FOLD retains a consistent advantage over pruning. The figure follows the same setup as Fig. 2 in the main paper, but for MAG2.

## D.3 VIT-B/32 ON CIFAR-10

The 200 Vision Transformers (ViT) also from (Andriushchenko et al., 2023), width=256, were trained on CIFAR-10, batch size 128, for 200 epochs with a cosine learning rate schedule and linear warmup. The main hyperparameters are summarized in Tab. 4. We made use of the maximum learning rate, the use of data augmentation, and the use of Sharpness-Aware Minimization (SAM) in our evaluations. All other settings were fixed.

| Parameter | Values |
|---|---|
| Optimizer | sgd |
| Max / Base Learning Rate (`lr_max`) | from 0.0504 to 4.9759 |
| SAM Strength (`sam_rho`) | 0.0, 0.05, 0.1 |
| Standard Augmentation (`augm`) | True, False |
| RandAugment (`randaug`) | True, False |

Table 3: Fixed and varying parameters for PreActResNet18 training on CIFAR-10.

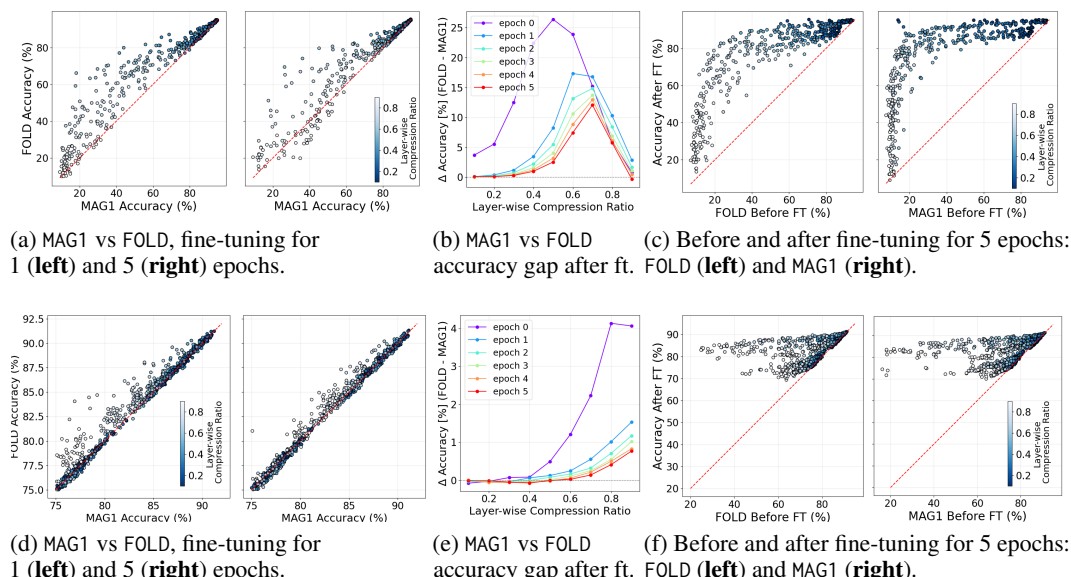

(a) MAG1 vs FOLD, fine-tuning for 1 (**left**) and 5 (**right**) epochs.

(b) MAG1 vs FOLD accuracy gap after ft.

(c) Before and after fine-tuning for 5 epochs: FOLD (**left**) and MAG1 (**right**).

(d) MAG1 vs FOLD, fine-tuning for 1 (**left**) and 5 (**right**) epochs.

(e) MAG1 vs FOLD accuracy gap after ft.

(f) Before and after fine-tuning for 5 epochs: FOLD (**left**) and MAG1 (**right**).

Figure 10: **`FOLD` outperforms `MAG1` after full fine-tuning for 1–5 epochs on PreActResNet18 and ViT-B/32 on CIFAR-10.** Results for PreActResNet18 (**top**) and ViT-B/32 (**bottom**). **(a,d)** accuracy of MAG1 vs. FOLD after 1 and 5 epochs of fine-tuning. **(b,e)** accuracy gap $\Delta$ over epochs, remaining positive. **(c,f)** accuracy trajectories from post-compression through 5 epochs, showing faster recovery and higher final accuracy for FOLD. The figure extends Fig. 3 in the main paper to PreActResNet18 and ViT-B/32 architectures where FOLD is benchmarked against MAG1.

### D.4 CLIP VIT-B/32 ON IMAGENET-1K

CLIP (Radford et al., 2021) models are known for the widespread use of CLIP features (Ramesh et al., 2022). We use the pool of models introduced by Wortsman et al. (2022b), who fine-tuned the CLIP ViT-B/32 architecture on ImageNet-1K multiple times using different randomly sampled training hyperparameters[7]. These hyperparameters include learning rate, number of training epochs, weight decay, label smoothing, and augmentation strategies, as stated in (Wortsman et al., 2022b). The resulting collection of 72 fine-tuned models provides a strong basis for evaluating the performance of model folding compared to pruning on CLIP ViT architectures. All checkpoints were evaluated jointly in our study, without parameter-specific ablations.

### D.5 LLAMA-60M ON COLOSSAL CLEAN CRAWLED CORPUS (C4)

We train 36 LLaMA-family models (Touvron et al., 2023a;b) with 60M and 130M parameters on the Colossal Clean Crawled Corpus (C4) (Raffel et al., 2020) on a NVIDIA DGX Station A100 featuring eight NVIDIA A100 GPUs (each equipped with 80GB memory). The training time for a LLaMA-60M model is about 45 minutes. Tab. 5 summarizes the fixed hyperparameters used to train LLaMA-60M and LLaMA-130M. The learning rate is linearly warmed up, followed by a cosine

---

[7]Download link: `https://github.com/mlfoundations/model-soups/releases/`

| Parameter | Values |
|---|---|
| Optimizer | sgd |
| Max / Base Learning Rate (`lr_max`) | from 0.005087 to 0.492936 |
| SAM Strength (`sam_rho`) | 0.0, 0.05, 0.1 |
| Standard Augmentation (`augm`) | True, False |
| RandAugment (`randaug`) | True, False |

Table 4: Fixed and varying parameters for ViT-B/32 Base training on CIFAR-10.

annealing schedule that decays to 10% of the initial value. We use the T5-base tokenizer (Raffel et al., 2023) and AdamW optimizer, consistent with prior work (Glentis et al., 2025; Han et al., 2024).

| Params | Hidden | Intermediate | Heads | Layers | Steps | Data (Tokens) |
|---|---|---|---|---|---|---|
| 60M | 512 | 1376 | 8 | 8 | 11K | 1.3B |
| 130M | 768 | 2048 | 12 | 12 | 22K | 2.6B |

Table 5: Training hyperparameters of LLaMA-60M architecture.

Note that in our work, pruning and folding are applied exclusively to the feed-forward network (FFN) layers of the trained LLaMA-60M and LLaMA-130M models.

## E    EXTENDED EMPIRICAL COMPARISON OF FOLDING AND PRUNING

We provide additional experiments to complement the main results. Fig. 8 mirrors the setup of Fig. 1 in the main paper, but replaces the L1 criterion for magnitude pruning with L2 (`MAG2`). Similarly, Fig. 9, Fig. 10, Fig. 11, and Fig. 12 extend the corresponding figures in the main paper to other network architectures and to the L2 case. Across all comparisons, the qualitative picture remains the same: `FOLD` consistently matches or outperforms magnitude pruning, independent of the chosen norm.

We further include ablations to study the robustness of these findings with respect to training hyperparameters. Fig. 13, Fig. 14, and Fig. 15 report the effect of varying learning rate, SAM strength, and RandAugment, respectively. Finally, Fig. 16 shows the influence of weight decay. Taken together, these studies confirm that the relative advantage of `FOLD` is stable across different regularization strategies and training configurations.

## F    ADDITIONAL ANALYSES: SHARPNESS, RUNTIME, AND LLMs

Below we report additional evaluations. We extend our study by training and compressing 60M- and 130M-parameter LLaMA models on C4, and provide analyses of sharpness, and measure runtime overhead.

### F.1    SHARPNESS AND BARRIER ANALYSIS

We compute sharpness following the implementation by Andriushchenko et al. (2023). Sharpness for CLIP is measured only on the final projection layer using $\sim$1000 images, while for PreActResNets it is computed over the full model and dataset.

Sharpness increases with compression ratio for all methods and architectures (Fig. 19), reaching a peak before stronger compression pushes the model out of its original basin and into a flatter, lower-capacity region. This rise–then–fall pattern appears consistently in both PreActResNet and CLIP models, see Fig. 19.

The correlation analysis in Fig. 20 further supports this interpretation. Across the 200 compressed ResNet18 models, 50 compressed PreActResNet18s and 72 compressed CLIP models, `FOLD` exhibits

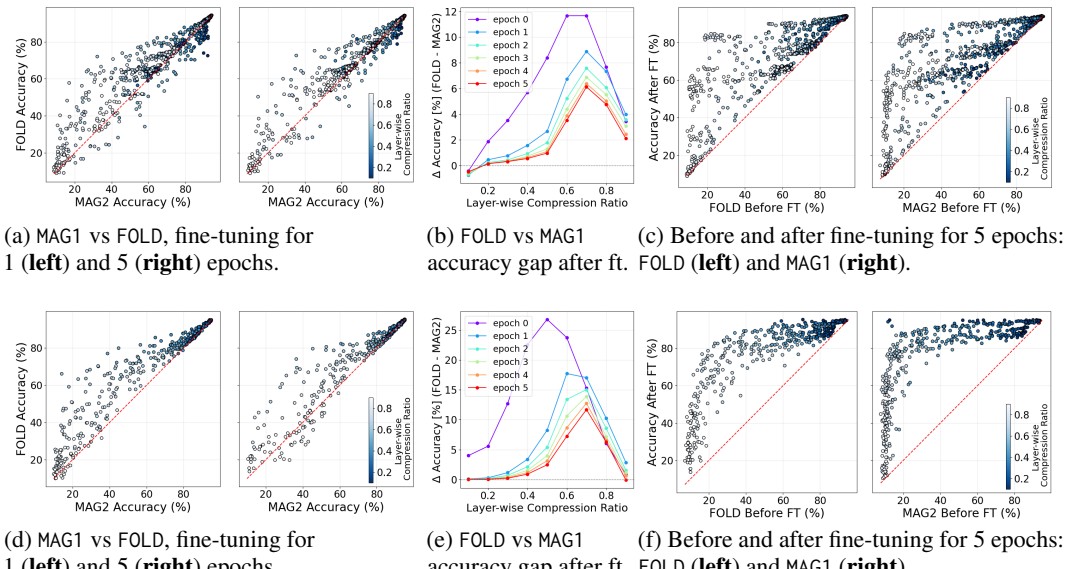

(a) MAG1 vs FOLD, fine-tuning for 1 (**left**) and 5 (**right**) epochs.

(b) FOLD vs MAG1 accuracy gap after ft.

(c) Before and after fine-tuning for 5 epochs: FOLD (**left**) and MAG1 (**right**).

(d) MAG1 vs FOLD, fine-tuning for 1 (**left**) and 5 (**right**) epochs.

(e) FOLD vs MAG1 accuracy gap after ft.

(f) Before and after fine-tuning for 5 epochs: FOLD (**left**) and MAG1 (**right**).

Figure 11: **Folded models retain their accuracy advantage after fine-tuning.** Results for ResNet18 trained by Adam (**top row**) and PreActResNet18 trained by SGD on CIFAR-10 (**bottom row**): **(a,d)** compares post-compression accuracy of magnitude pruning with L2 criterion (MAG2) versus folding (FOLD) after 1 and 5 epochs of fine-tuning. **(b,e)** show the accuracy gap between folding and pruning as a function of fine-tuning epochs, demonstrating that folding maintains a consistent lead, *i.e.*, the FOLD accuracy delta is positive. **(c,f)** illustrate accuracy trajectories before and after 5 epochs of fine-tuning for both methods, highlighting that folded models recover accuracy faster and reach higher final performance than pruned models. The figure extends Fig. 3 in the main paper and Fig. 10 in the appendix to MAG2.

| weight_decay | warmup_steps | max_lr | PPL↓ 0% sparsity | PPL↓ MAG2 (20%) | PPL↓ FOLD (20%) | PPL↓ MAG2 (50%) | PPL↓ FOLD (50%) |
|---|---|---|---|---|---|---|---|
| 0.01 | 1100 | 0.001 | 23.90 | 39.88 | **39.48** | **236.16** | 308.77 |
| 0.01 | 2200 | 0.001 | 23.99 | **38.75** | 39.61 | **259.79** | 469.25 |
| 0.01 | 3300 | 0.001 | 24.08 | **38.54** | 39.10 | **289.67** | 451.27 |
| 0.0 | 1100 | 0.001 | 24.01 | 42.39 | **42.27** | **270.31** | 477.70 |
| 0.0 | 2200 | 0.001 | 24.12 | **40.01** | 41.53 | **239.25** | 489.48 |
| 0.0 | 3300 | 0.001 | 24.19 | **38.72** | 40.31 | **277.10** | 531.09 |
| 0.01 | 1100 | 0.005 | 42.11 | 72.31 | **62.63** | 536.53 | **298.65** |
| 0.01 | 2200 | 0.005 | 22.82 | 52.18 | **40.46** | 824.69 | **333.59** |
| 0.01 | 3300 | 0.005 | 22.66 | 44.35 | **36.22** | 589.33 | **222.21** |
| 0.0 | 1100 | 0.005 | 44.92 | 73.38 | **63.75** | 414.17 | **261.23** |
| 0.0 | 2200 | 0.005 | 23.32 | 57.62 | **43.04** | 1616.74 | **342.64** |
| 0.0 | 3300 | 0.005 | 23.00 | 46.87 | **39.11** | 904.07 | **305.85** |
| 0.01 | 1100 | 0.01 | 300.95 | 302.26 | **301.87** | 401.28 | **361.10** |
| 0.01 | 2200 | 0.01 | 66.48 | 88.09 | **84.30** | 398.16 | **252.99** |
| 0.01 | 3300 | 0.01 | 54.34 | 97.35 | **76.15** | 440.71 | **229.42** |
| 0.0 | 1100 | 0.01 | 282.11 | 282.48 | **282.38** | 345.74 | **329.58** |
| 0.0 | 2200 | 0.01 | 140.20 | 169.78 | **149.58** | 352.14 | **234.69** |
| 0.0 | 3300 | 0.01 | 86.18 | 118.05 | **100.14** | 339.37 | **179.43** |

Table 6: **Evaluation of FOLD and MAG2 on LLaMA-130M** (in addition to LLaMA-60M evaluations in Tab. 1). We train and evaluate 18 LLaMA-family models with 130M parameters on C4 while varying max_lr, warmup steps, and weight decay. Columns show perplexity of the pretrained model (0% sparsity) and perplexity after structured magnitude pruning and folding with 20% and 50% sparsity in FFN blocks. For higher learning rates, especially for the settings with the best achieved performance in each sparsity category (underlined), FOLD consistently outperforms MAG2 (bold).

negative correlations between $\Delta$-sharpness and $\Delta$-accuracy ($\Delta$Accuracy $= \text{Acc}(\text{FOLD}) - \text{Acc}(\text{MAG})$). As shown in the scatter plots and correlation tables of Fig. 20, larger reductions in sharpness under FOLD relative to MAG are associated with larger accuracy gains. This relationship holds across the

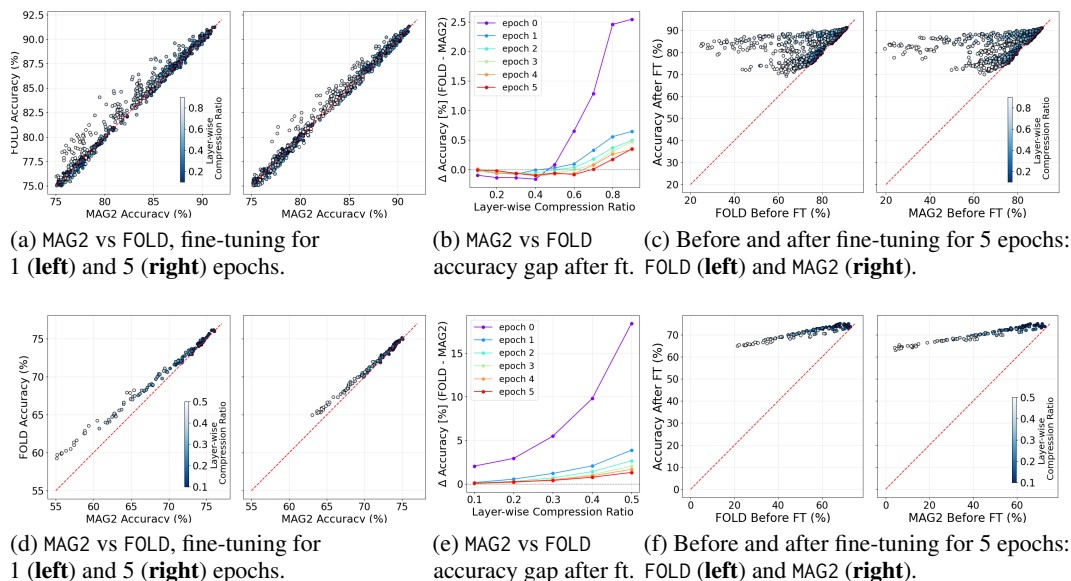

(a) MAG2 vs FOLD, fine-tuning for 1 (**left**) and 5 (**right**) epochs.

(b) MAG2 vs FOLD accuracy gap after ft.

(c) Before and after fine-tuning for 5 epochs: FOLD (**left**) and MAG2 (**right**).

(d) MAG2 vs FOLD, fine-tuning for 1 (**left**) and 5 (**right**) epochs.

(e) MAG2 vs FOLD accuracy gap after ft.

(f) Before and after fine-tuning for 5 epochs: FOLD (**left**) and MAG2 (**right**).

Figure 12: **FOLD outperforms MAG2 after full fine-tuning for 1–5 epochs on ViT-B/32 and CLIP ViT-B/32.** Results for ViT-B/32 on CIFAR-10 (**top**) and CLIP ViT-B/32 on ImageNet-1K (**bottom**). (**a,d**) accuracy of MAG2 vs. FOLD after 1 and 5 epochs of fine-tuning. (**b,e**) accuracy gap $\Delta$ over epochs, remaining positive. (**c,f**) accuracy trajectories from post-compression through 5 epochs, showing faster recovery and higher final accuracy for FOLD. The figure extends Fig. 3 in the main paper and Fig. 10 in the appendix to MAG2.

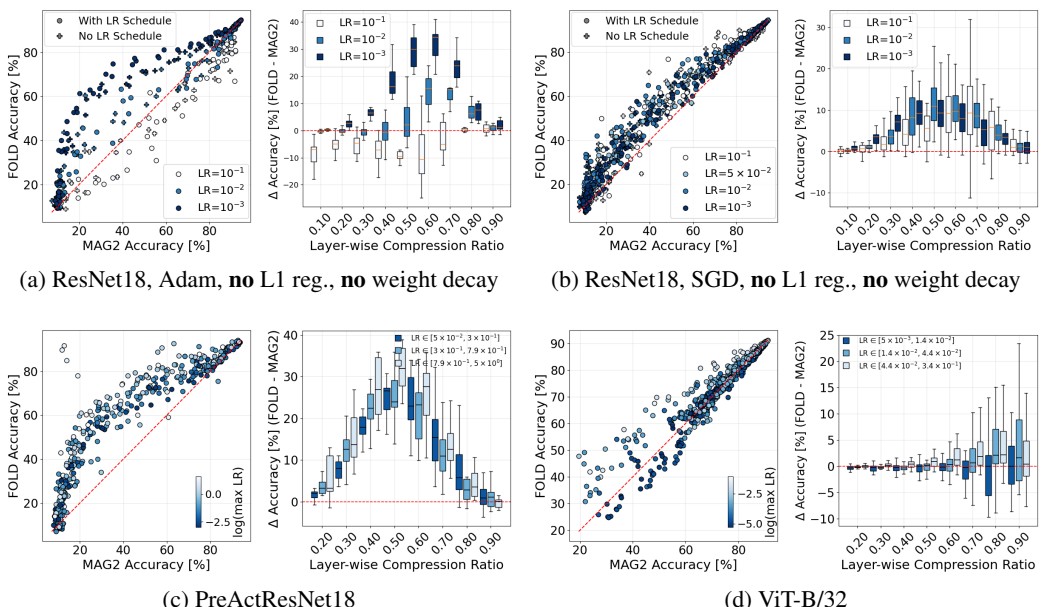

(a) ResNet18, Adam, **no** L1 reg., **no** weight decay

(b) ResNet18, SGD, **no** L1 reg., **no** weight decay

(c) PreActResNet18

(d) ViT-B/32

Figure 13: **Learning rate modulates folding's edge.** Post-compression accuracy of MAG2 and FOLD across learning rates: ResNet18 with Adam (**a**) and SGD (**b**), PreActResNet18 (**c**), and ViT-B/32 (**d**). FOLD typically leads at moderate–low rates; the gap shrinks or reverses at very high rates, and closes again at extremely small rates. The same setup as in Fig. 4 in the main paper, but for MAG2.

evaluated compression ratios, up to the point where one of the models leaves the original basin and sharpness becomes less informative.

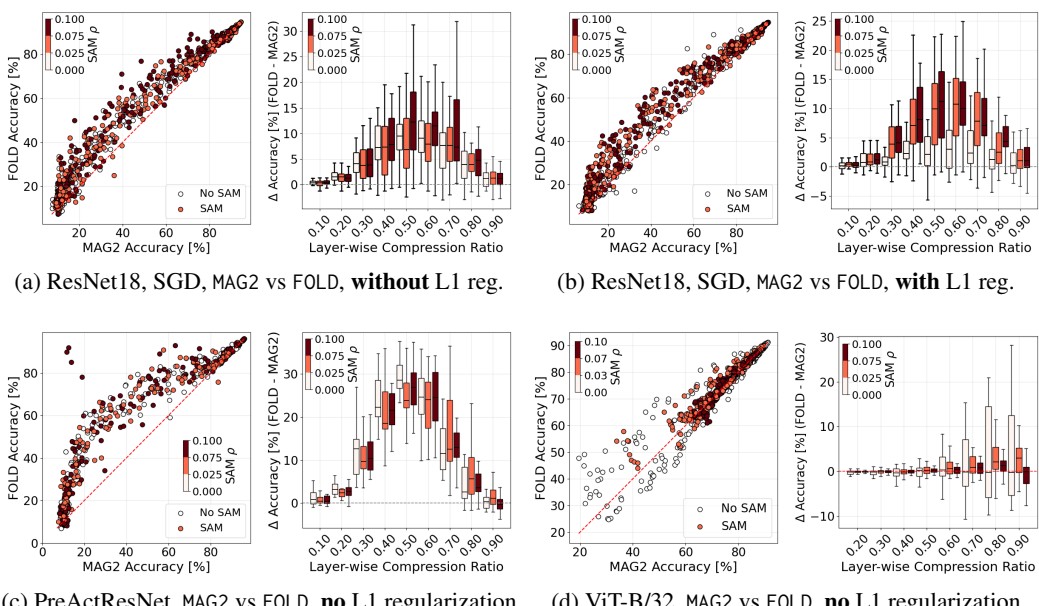

(a) ResNet18, SGD, MAG2 vs FOLD, **without** L1 reg.

(b) ResNet18, SGD, MAG2 vs FOLD, **with** L1 reg.

(c) PreActResNet, MAG2 vs FOLD, **no** L1 regularization

(d) ViT-B/32, MAG2 vs FOLD, **no** L1 regularization

Figure 14: **SAM can boost model compression.** Post-compression accuracy under training with / without SAM. **(a)** ResNet18 (Adam), no L1. **(b)** ResNet18 (Adam), L1= $10^{-5}$. **(c)** PreActResNet18 (SGD), no L1. **(d)** ViT-B/32, no L1. The figure extends the results in Fig. 6 to MAG2.

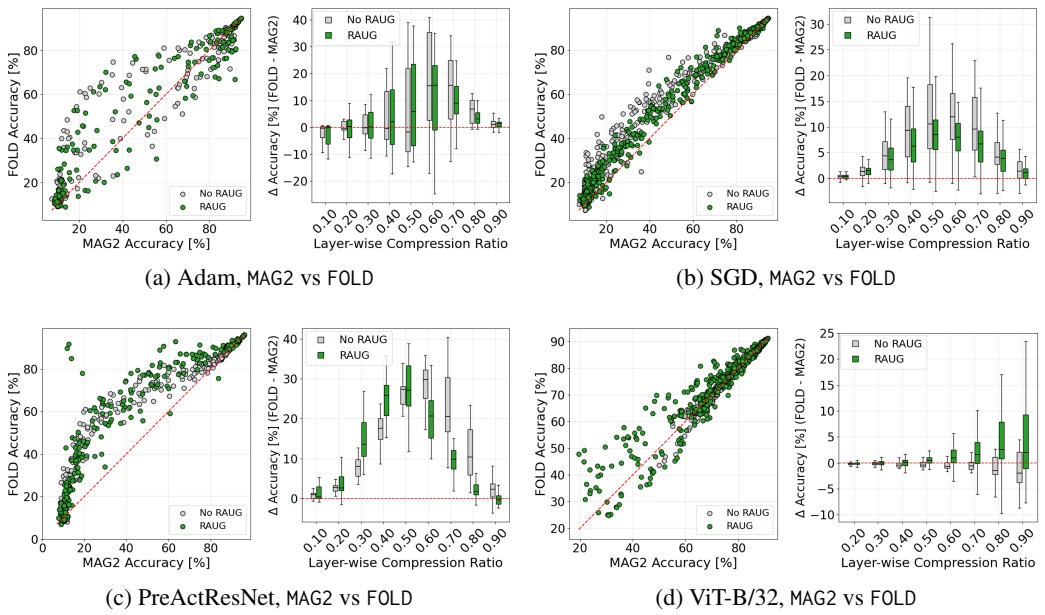

(a) Adam, MAG2 vs FOLD

(b) SGD, MAG2 vs FOLD

(c) PreActResNet, MAG2 vs FOLD

(d) ViT-B/32, MAG2 vs FOLD

Figure 15: **Random augmentations narrow the folding–pruning gap.** Post-compression accuracy on ResNet18 (CIFAR-10) trained without vs. with random augmentations: **(a)** Adam, **(b)** SGD, **(c)** PreActResNet, **(d)** ViT-B/32. The figure extends Fig. 7 to MAG2.

In addition to the global sharpness trends discussed above, Fig. 21 and Fig. 22 provide a more fine-grained view of how training hyperparameters influence the relationship between Δ-sharpness and Δ-accuracy under compression. For Adam-trained ResNet models (Fig. 21), the scatter plots reveal a strong and stable negative correlation: whenever FOLD produces lower sharpness than magnitude pruning, it also achieves higher accuracy across almost all pruning ratios. The structure of the point clouds, especially at high learning rates, shows that Adam's adaptive scaling can induce

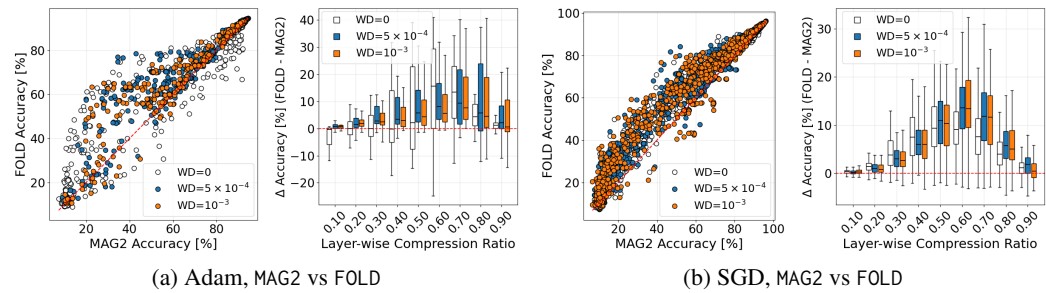

(a) Adam, MAG2 vs FOLD         (b) SGD, MAG2 vs FOLD

Figure 16: **ResNet18: Weight Decay.** Test accuracy of ResNet18 checkpoints trained with varying weight decay values. Weight decay does not diminish the advantage of FOLD compared to MAG2, especially for SGD-trained models.

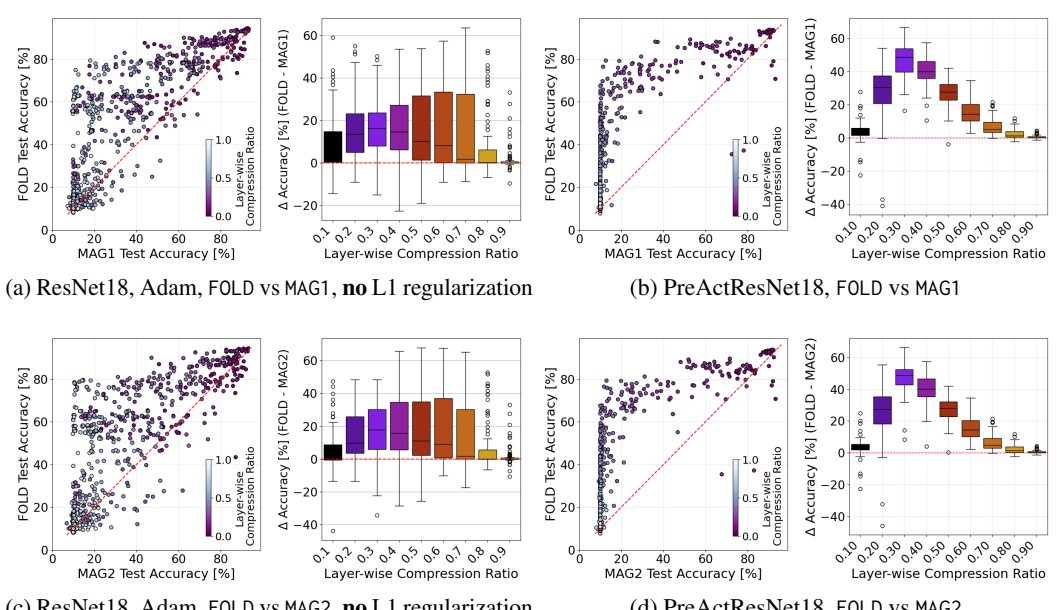

(a) ResNet18, Adam, FOLD vs MAG1, **no** L1 regularization      (b) PreActResNet18, FOLD vs MAG1

(c) ResNet18, Adam, FOLD vs MAG2, **no** L1 regularization      (d) PreActResNet18, FOLD vs MAG2

Figure 17: **Folding vs. magnitude pruning before REPAIR.** The same setup as in Fig. 1 and Fig. 8 for CNNs (ResNet18 and PreActResNet18 on CIFAR-10), but the performance is compared for both pruning and folding before REPAIR. **Top row:** MAG1, **bottom row:** MAG2. In both cases, folding shows stronger performance already before data-based REPAIR is applied.

highly anisotropic sharpness profiles, which in turn amplify the divergence between the compression trajectories of FOLD and MAG.

In contrast, SGD-trained models (Fig. 22) exhibit a weaker and more dispersed relationship between $\Delta$-sharpness and $\Delta$-accuracy, consistent with the flatter and more isotropic minima typically found by SGD. Under SGD, FOLD often remains slightly flatter than magnitude pruning even when $\Delta$-sharpness $\approx 0$, explaining why FOLD maintains a mild yet more weakly correlated accuracy advantage. The interaction with SAM and augmentation further differs across optimizers: SAM tightens the $\Delta$-sharpness distribution under SGD, stabilizing the performance gap in favor of FOLD, while RandAug tends primarily to reduce variance without introducing strong directional trends.

These results highlight that the predictive power of sharpness for pruning outcomes is optimizer- and hyperparameter-dependent: sharpness differences are highly informative for Adam-trained networks but less so for SGD, even though FOLD consistently follows a smoother and less disruptive compression path than magnitude-based pruning in both regimes.

These findings align with recent work linking compression and landscape geometry. AdaSAP (Bair et al., 2024) treats pruning as a sharpness-aware process, and Zhang et al. (2025) show that

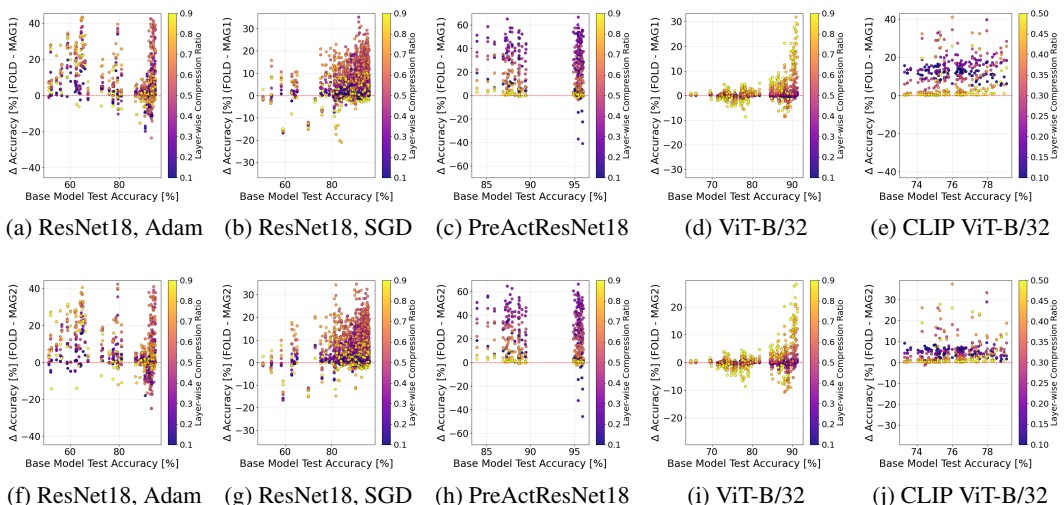

Figure 18: **Uncompressed model accuracy vs. performance difference** $\Delta$**Accuracy** $=$ Acc(**FOLD**) $-$ Acc(**MAG**)**.** The same setup as in Fig. 1 and Fig. 8. **Top row:** MAG1, **Bottom row:** MAG2. Model folding shows strong performance on models of different quality, with amplified effect on high-performing models (especially on ResNet18, SGD and ViT-B/32).

feasible pruning ratios depend on intrinsic flatness. Our results support this perspective: compression initially increases sharpness as degrees of freedom are removed within the same basin, but stronger compression forces the model into a flatter basin with reduced curvature. FOLD follows this trajectory more smoothly, maintaining basin structure and yielding lower barriers than MAG.

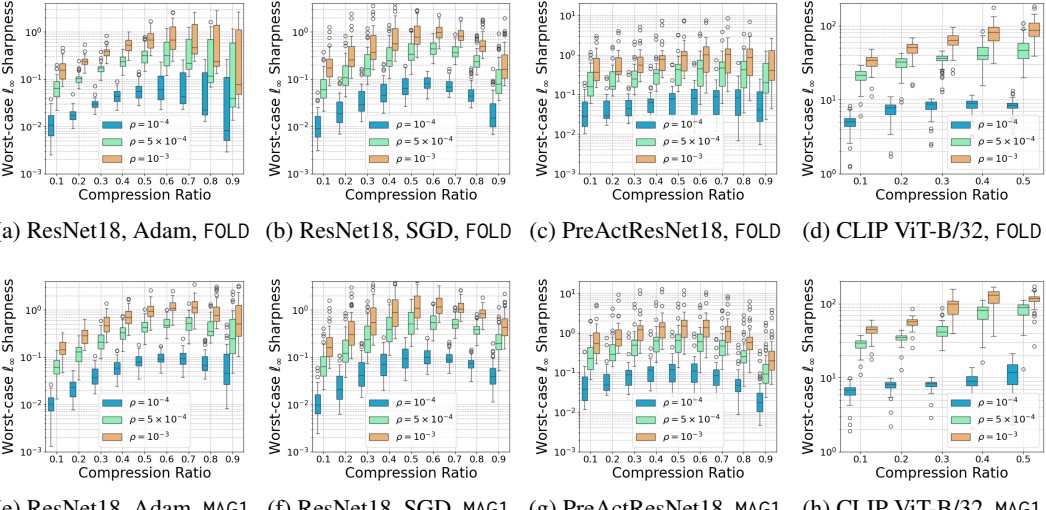

Figure 19: **Worst-case $\ell_\infty$ sharpness as a function of compression ratio across architectures and pruning methods.** Each subplot reports the sharpness distribution for independently trained models at three perturbation radii ($\rho = 10^{-4}$, $5 \times 10^{-4}$, $10^{-3}$). Panels (a)–(d) show FOLD sharpness for ResNet18 trained with Adam and SGD, PreActResNet18, and CLIP ViT-B/32, respectively. Panels (e)–(h) show the corresponding results for MAG1. Observed trends: (i) Sharpness generally increases with compression ratio up to moderate levels before flattening or dropping at extreme compression. (ii) FOLD produces on-average lower sharpness than MAG1. (iii) Transformer models (CLIP ViT-B/32) experience substantially sharper solutions under compression compared to residual networks. These patterns indicate that FOLD maintains flatter loss landscapes across a wide range of settings, whereas MAG more often drives models toward sharper and less stable minima.

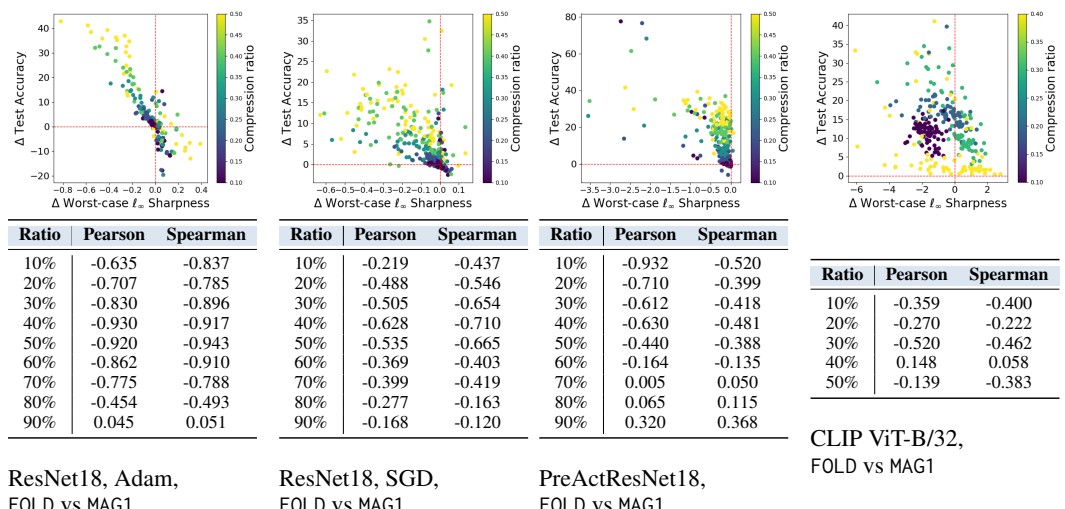

| Ratio | Pearson | Spearman |
|-------|---------|----------|
| 10% | -0.635 | -0.837 |
| 20% | -0.707 | -0.785 |
| 30% | -0.830 | -0.896 |
| 40% | -0.930 | -0.917 |
| 50% | -0.920 | -0.943 |
| 60% | -0.862 | -0.910 |
| 70% | -0.775 | -0.788 |
| 80% | -0.454 | -0.493 |
| 90% | 0.045 | 0.051 |

ResNet18, Adam,
FOLD vs MAG1

| Ratio | Pearson | Spearman |
|-------|---------|----------|
| 10% | -0.219 | -0.437 |
| 20% | -0.488 | -0.546 |
| 30% | -0.505 | -0.654 |
| 40% | -0.628 | -0.710 |
| 50% | -0.535 | -0.665 |
| 60% | -0.369 | -0.403 |
| 70% | -0.399 | -0.419 |
| 80% | -0.277 | -0.163 |
| 90% | -0.168 | -0.120 |

ResNet18, SGD,
FOLD vs MAG1

| Ratio | Pearson | Spearman |
|-------|---------|----------|
| 10% | -0.932 | -0.520 |
| 20% | -0.710 | -0.399 |
| 30% | -0.612 | -0.418 |
| 40% | -0.630 | -0.481 |
| 50% | -0.440 | -0.388 |
| 60% | -0.164 | -0.135 |
| 70% | 0.005 | 0.050 |
| 80% | 0.065 | 0.115 |
| 90% | 0.320 | 0.368 |

PreActResNet18,
FOLD vs MAG1

| Ratio | Pearson | Spearman |
|-------|---------|----------|
| 10% | -0.359 | -0.400 |
| 20% | -0.270 | -0.222 |
| 30% | -0.520 | -0.462 |
| 40% | 0.148 | 0.058 |
| 50% | -0.139 | -0.383 |

CLIP ViT-B/32,
FOLD vs MAG1

Figure 20: **Correlation between $\Delta$-sharpness and $\Delta$-accuracy across architectures and pruning baselines.** Each column shows the relationship between pruning–induced differences in worst-case $\ell_\infty$ sharpness ($\Delta$ sharpness = FOLD − MAG) and differences in test accuracy ($\Delta$ accuracy = FOLD − MAG), for the pair of pruning methods indicated below each plot. Color encodes the layer-wise compression ratio. The tables underneath each subplot report Pearson and Spearman correlations at every compression ratio, quantifying how predictive the sharpness difference is of the accuracy difference. Results are shown for FOLD vs MAG1 for ResNet18 trained with Adam and SGD, PreActResNet18 and CLIP ViT-B/32. Statistics are computed over 200 independently trained ResNets18 and 50 PreActResNet18 on CIFAR-10, and 72 CLIP ViT-B/32 models on ImageNet-1K. Correlations use the sharpness value at $\rho = 5 \times 10^{-4}$. Results at $\rho = 10^{-4}$ and $\rho = 10^{-3}$ are qualitatively very close.

## F.2 RUNTIME OVERHEAD AND EQUIVALENCE OF COMPRESSED MODELS

We profile both the compression procedures and the inference behavior of the resulting compressed models on a dedicated DGX A100 server equipped with dual-socket AMD EPYC 7742 CPUs (256 hardware threads) and 8× NVIDIA A100 80GB GPUs. All measurements use the THOP profiler[8] and report compression time, peak memory during compression, per-batch latency, FLOPs, and peak forward-pass memory before and after compression. For each architecture (PreActResNet18 and CLIP ViT-B/32), all compression methods generate the *same* compressed network topology (identical channel counts and tensor shapes). Consequently, all methods yield identical FLOPs and nearly identical latencies, demonstrating that inference-time behavior is determined entirely by the resulting architecture, not by the choice of compression algorithm. FOLD introduces a moderate one-off compression overhead, but its inference-time profile matches the other compressed models.

| Method | Params | Comp. time [s] | Comp. peak mem [MB] | Lat. [ms/batch] | Lat. [ms/img] | FLOPs [MFLOPs/img] | Fwd peak mem [MB] |
|--------|--------|----------------|---------------------|-----------------|---------------|--------------------|--------------------| 
| Original | 11,172,170 | – | – | 3.69 | 0.0288 | 557.65 | 214.30 |
| FOLD | 4,008,346 | 9.48 | 157.47 | 3.17 | 0.0248 | 199.05 | 170.53 |
| MAG | 4,008,346 | 1.77 | 115.22 | 3.15 | 0.0246 | 199.05 | 169.82 |

Table 7: Runtime characteristics of PreActResNet18 before and after compression (64.1% parameter reduction). Latency is measured for a full batch. FLOPs are reported per image. Comp. time and comp. peak mem refer to the overhead of running the compression method once.

Table 7 compares the original and compressed PreActResNet18 at a layer-wise compression ratio of 0.4 (*i.e.*, a 64.1% reduction in model parameters). Compression reduces FLOPs from 557.65 MFLOPs/image to 199.05 MFLOPs/image (a 64.3% reduction), improves latency from 3.69 ms/batch to roughly 3.15 ms/batch, and lower peak forward-pass memory (from 214.30 MB to about 170 MB).

---

[8]https://github.com/ultralytics/thop

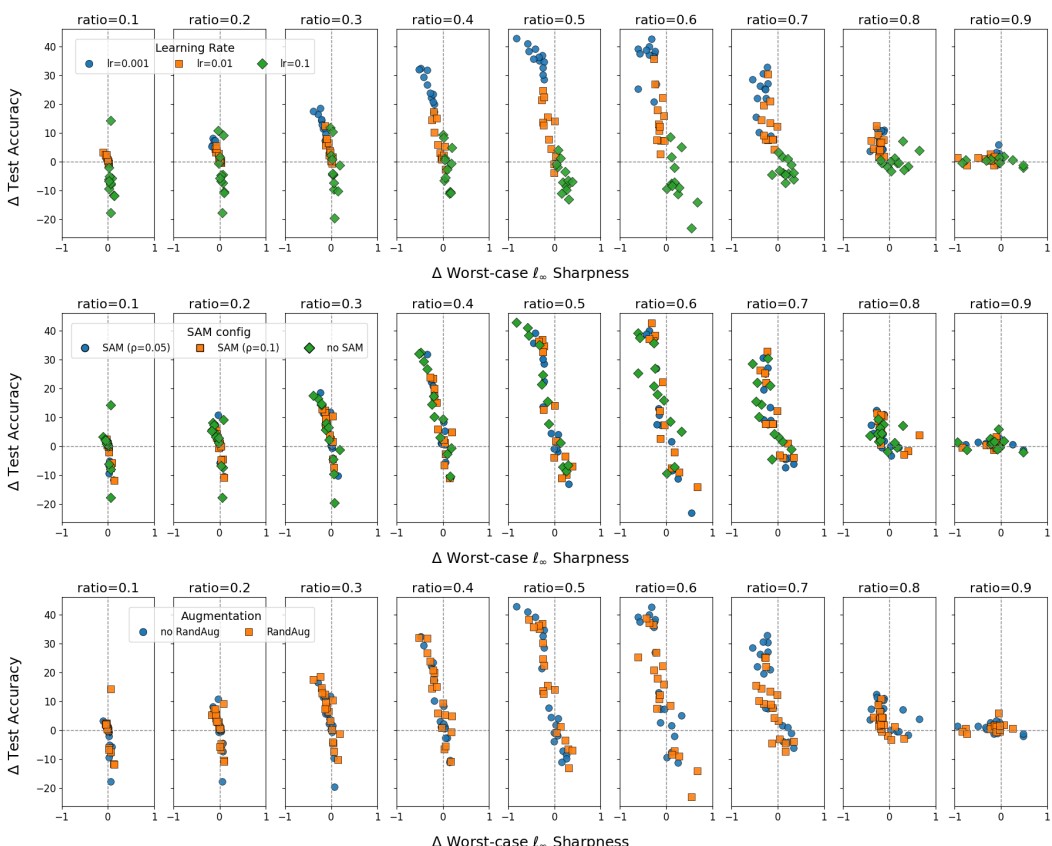

Figure 21: **Sharpness-accuracy trade-off between `FOLD` and `MAG1` for ResNet18 trained with Adam.** Each column corresponds to a layer-wise compression ratio (0.1–0.9), and the three rows group models by learning rate, SAM configuration (including $\rho$), and RandAug usage. Points show $\Delta$ worst-case $\ell_\infty$ sharpness (`FOLD` − `MAG1`) vs. $\Delta$ test accuracy (`FOLD` − `MAG1`). **Observations:** (1) The difference in model sharpness strongly predicts the difference in performance between `FOLD` and `MAG1` across almost all pruning ratios. (2) *Learning rate:* higher learning rates lead to more dispersed sharpness changes. For models trained with Adam using high learning rates, `MAG1` moves the model along a less sharp path compared to `FOLD`, which struggles to catch up. However, the behavior flips for moderate and low learning rates. (3) *SAM/$\rho$:* using SAM reduces the variability in the sharpness shift between methods, especially for larger $\rho$. $\Delta$ sharpness gets closer to zero. (4) *RandAug:* augmentation show little specific visible trend.

| Method | Params | Comp. time [s] | Comp. peak mem [MB] | Lat. [ms/batch] | Lat. [ms/img] | FLOPs [MFLOPs/img] | Fwd peak mem [MB] |
|---|---|---|---|---|---|---|---|
| Original | 151,790,313 | – | – | 19.851 | 0.6203 | 2946.76 | 684.18 |
| FOLD | 140,447,253 | 92.833 | 681.23 | 17.343 | 0.5420 | 2379.98 | 636.97 |
| MAG | 140,447,253 | 2.627 | 625.61 | 17.372 | 0.5429 | 2379.98 | 637.88 |

Table 8: Runtime characteristics of CLIP ViT-B/32 before and after compression (7.47% parameter reduction). Latency is measured for a full batch. FLOPs are reported per image. Comp. time and Comp. peak mem refer to the one-off overhead of running the compression method.

Table 8 shows the same evaluation for CLIP ViT-B/32, where FFN blocks are compressed to with 20% layer-wise compression ratio. Here, FLOPs decrease from 2946.76 MFLOPs/image to 2379.98 MFLOPs/image (a 19.2% reduction), and latency improves from 19.8 ms/batch to roughly 17.35 ms/batch (about $1.14\times$ speed-up). Again, `FOLD` is the slowest method due to its iterative nature to compute $k$-means clusters, but the compressed models share the same FLOPs, memory and latency.

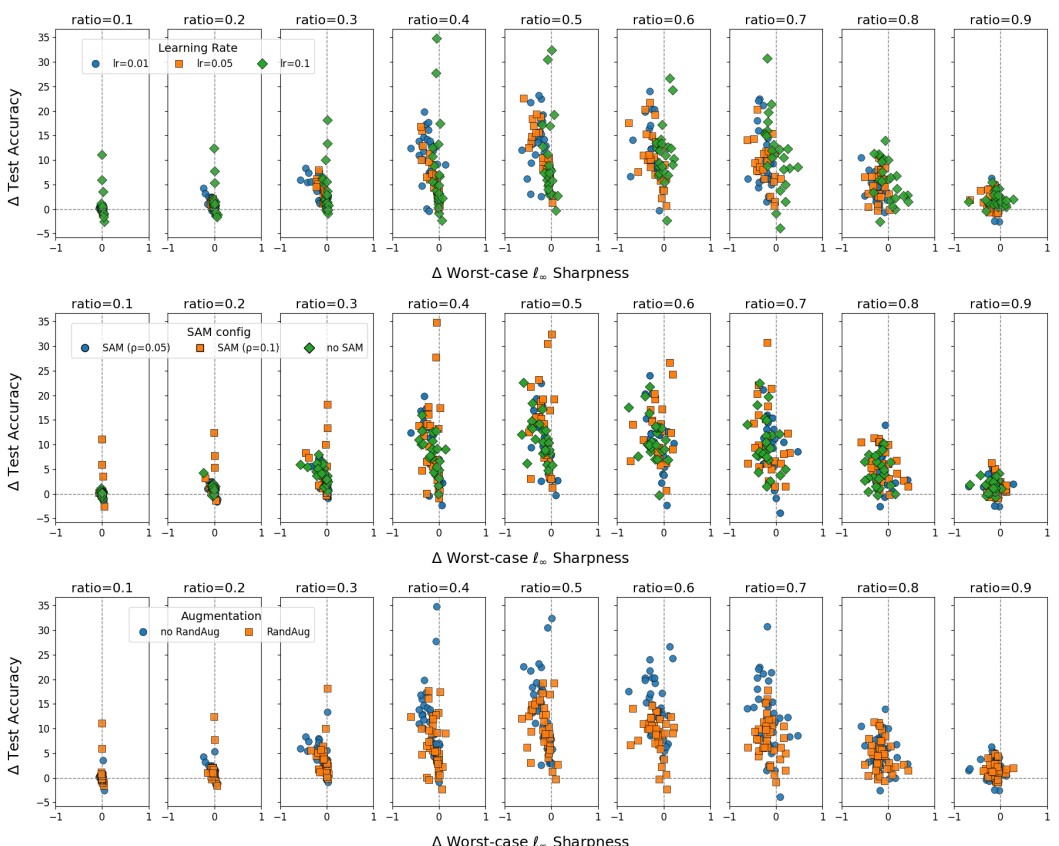

Figure 22: **Sharpness-accuracy trade-off between `FOLD` and `MAG1` for ResNet18 trained with SGD.** Each column corresponds to a layer-wise compression ratio (0.1–0.9), and the three rows group models by learning rate, SAM configuration (including $\rho$), and RandAug usage. Points show $\Delta$ worst-case $\ell_\infty$ sharpness (`FOLD` − `MAG1`) vs. $\Delta$ test accuracy (`FOLD` − `MAG1`). **Observations:** (1) For SGD, the relationship between $\Delta$ sharpness and $\Delta$ accuracy is visibly weaker and more scattered than for Adam, reflecting the flatter and more anisotropic minima found by SGD. (2) *Learning rate:* Unlike Adam, higher SGD learning rates do not systematically increase the sharpness gap between the methods. For most ratios, `FOLD` tends to remain less sharp than `MAG1`, producing positive $\Delta$ accuracy even when $\Delta$ sharpness is near zero. (3) *SAM/$\rho$:* SAM has a strong flattening effect under SGD—$\Delta$ sharpness clusters tightly around zero, and the accuracy advantage of `FOLD` becomes more stable as $\rho$ increases. (4) *RandAug:* Augmentation increases robustness to pruning under SGD, reducing the spread in $\Delta$ accuracy and further weakening the sharpness–accuracy link. Overall, SGD-trained models exhibit a regime where `FOLD` consistently follows a gentler sharpness trajectory than `MAG1`, leading to a clearer accuracy advantage at moderate pruning ratios.

## F.3 IMPACT OF ONE RANK SLACK AND SINGLETON FOLDING

In this section we separate two effects in our pruning vs. folding comparison: (i) the influence of the one-rank slack in Theorems 2.1–2.2, and (ii) the intrinsic difference between pruning and folding as projection operators. We therefore contrast the gain from increasing the pruning rank from $k$ to $k+1$ (blue curves in Fig. 23) with the gain from replacing pruning by folding at the same nominal rank (orange curves). This isolates the contribution of rank from the contribution of the projection geometry.

Fig. 23 shows both effects: for each weight matrix $\mathbf{W}$ (in every layer of ViT or ResNet18), it plots the relative Frobenius error change when the retained rank increases by one (blue) versus when the method changes from pruning to folding at fixed rank (orange), as a function of $k$.

| Layer | Params_fold | Params_mag | $\Delta p$ | FLOPs_fold | FLOPs_mag | $\Delta F$ | Act_fold | Act_mag | $\Delta a$ | NZ_fold | NZ_mag | $\Delta nz$ |
|---|---|---|---|---|---|---|---|---|---|---|---|---|
| conv1 | 1026 | 1026 | 0 | 1050624 | 1050624 | 0 | 38912 | 38912 | 0 | 38912 | 38912 | 0 |
| layer1.0.conv1 | 12996 | 12996 | 0 | 13307904 | 13307904 | 0 | 38912 | 38912 | 0 | 38912 | 38912 | 0 |
| layer1.0.conv2 | 12996 | 12996 | 0 | 13307904 | 13307904 | 0 | 38912 | 38912 | 0 | 38912 | 38912 | 0 |
| layer1.1.conv1 | 12996 | 12996 | 0 | 13307904 | 13307904 | 0 | 38912 | 38912 | 0 | 38912 | 38912 | 0 |
| layer1.1.conv2 | 12996 | 12996 | 0 | 13307904 | 13307904 | 0 | 38912 | 38912 | 0 | 38912 | 38912 | 0 |
| layer2.0.conv1 | 25992 | 25992 | 0 | 6653952 | 6653952 | 0 | 19456 | 19456 | 0 | 19456 | 19456 | 0 |
| layer2.0.conv2 | 51984 | 51984 | 0 | 13307904 | 13307904 | 0 | 19456 | 19456 | 0 | 19456 | 19456 | 0 |
| layer2.0.shortcut.0 | 2888 | 2888 | 0 | 739328 | 739328 | 0 | 19456 | 19456 | 0 | 19456 | 19456 | 0 |
| layer2.1.conv1 | 51984 | 51984 | 0 | 13307904 | 13307904 | 0 | 19456 | 19456 | 0 | 19456 | 19456 | 0 |
| layer2.1.conv2 | 51984 | 51984 | 0 | 13307904 | 13307904 | 0 | 19456 | 19456 | 0 | 19456 | 19456 | 0 |
| layer3.0.conv1 | 104652 | 104652 | 0 | 6697728 | 6697728 | 0 | 9792 | 9792 | 0 | 9792 | 9792 | 0 |
| layer3.0.conv2 | 210681 | 210681 | 0 | 13483584 | 13483584 | 0 | 9792 | 9792 | 0 | 9792 | 9792 | 0 |
| layer3.0.shortcut.0 | 11628 | 11628 | 0 | 744192 | 744192 | 0 | 9792 | 9792 | 0 | 9792 | 9792 | 0 |
| layer3.1.conv1 | 210681 | 210681 | 0 | 13483584 | 13483584 | 0 | 9792 | 9792 | 0 | 9792 | 9792 | 0 |
| layer3.1.conv2 | 210681 | 210681 | 0 | 13483584 | 13483584 | 0 | 9792 | 9792 | 0 | 9792 | 9792 | 0 |
| layer4.0.conv1 | 422739 | 422739 | 0 | 6763824 | 6763824 | 0 | 4912 | 4912 | 0 | 4912 | 4912 | 0 |
| layer4.0.conv2 | 848241 | 848241 | 0 | 13571856 | 13571856 | 0 | 4912 | 4912 | 0 | 4912 | 4912 | 0 |
| layer4.0.shortcut.0 | 46971 | 46971 | 0 | 751536 | 751536 | 0 | 4912 | 4912 | 0 | 4912 | 4912 | 0 |
| layer4.1.conv1 | 848241 | 848241 | 0 | 13571856 | 13571856 | 0 | 4912 | 4912 | 0 | 4912 | 4912 | 0 |
| layer4.1.conv2 | 848241 | 848241 | 0 | 13571856 | 13571856 | 0 | 4912 | 4912 | 0 | 4912 | 4912 | 0 |
| linear | 3080 | 3080 | 0 | 3070 | 3070 | 0 | 10 | 10 | 0 | 10 | 10 | 0 |
| TOTALS | 4003678 | 4003678 | 0 | 197725902 | 197725902 | 0 | 365370 | 365370 | 0 | 365370 | 365370 | 0 |

Table 9: Per-layer comparison of PreActResNet18 after `FOLD` and `MAG2` at compression ratio $0.4$. For each convolutional and linear layer we report parameters, per-image FLOPs, activation size, and the number of non-zero activations (effective activations). All per-layer differences are zero, confirming that parameters, FLOPs, activations, and effective activations are exactly matched between the two compressed models.

$$\Delta_{\text{rank}}(k) = \frac{\|\mathbf{W} - \mathbf{W}_p^{(k)}\|_F - \|\mathbf{W} - \mathbf{W}_p^{(k+1)}\|_F}{\|\mathbf{W}\|_F}.$$

This quantity measures the improvement obtained when increasing the retained rank from $k$ to $k+1$ within magnitude pruning `MAG2`. It isolates the rank slack effect. Across all examined layers, the improvement from a single additional retained channel is small, especially in deeper layers.

$$\Delta_{\text{method}}(k) = \frac{\|\mathbf{W} - \mathbf{W}_p^{(k)}\|_F - \|\mathbf{W} - \mathbf{W}_f^{\star(k)}\|_F}{\|\mathbf{W}\|_F}.$$

This measures the gain obtained by switching from structured magnitude pruning to optimal folding at the same rank $k$. The improvements are one to two orders of magnitude larger than the corresponding $\Delta_{\text{rank}}$ values for nearly all layers.

The results empirically support the clarification presented in the rebuttal: although Theorems 2.1-2.2 compare pruning at rank $k$ to folding at rank $k+1$, the contribution of the rank difference is negligible in practice. The blue curves show that $\|\mathbf{W} - \mathbf{W}_p^{(k)}\|_F$ changes only minimally when $k \to k+1$, while the orange curves demonstrate that folding provides a substantially tighter approximation than pruning at comparable compression levels. This confirms that the practical advantage of folding arises primarily from the richer family of cluster-based projections rather than the added rank.

Fig. 24 reports the relative squared error for pruning, special folding $\mathbf{W}_f'$, and optimal folding $\mathbf{W}_f^*$ across all layers of ResNet18 and ViT-B/32 on CIFAR10, evaluated at multiple keep ratios. Three consistent phenomena appear:

- **Pruning always yields the largest error.** For every layer and every keep ratio, the pruning curves lie above both folding curves. This confirms empirically that pruning introduces the largest distortion of the original weight matrix.

- **Special folding $\mathbf{W}_f'$ strictly improves over pruning.** The construction used in Theorem 2.1, obtained by merging all pruned rows, leads to smaller reconstruction error for all layers and all keep ratios. This empirically validates the first inequality $\text{error}(\mathbf{W}_p) \geq \text{error}(\mathbf{W}_f')$.

- **Optimal folding $\mathbf{W}_f^*$ achieves the smallest error.** The $k$-means solution consistently attains the lowest error, verifying the second inequality $\text{error}(\mathbf{W}_f') \geq \text{error}(\mathbf{W}_f^*)$.

Importantly, the gap between pruning and both folding methods is larger than the very small difference induced by adding a single additional cluster (*i.e.*, increasing the rank from $k$ to $k + 1$). This supports

the clarification made in the rebuttal: the practical advantage of folding does not stem from the $+1$ change in rank, but from the richer family of cluster-based projections that folding can realize.

The observed ordering, *i.e.*, $\mathrm{error}(\mathbf{W}_p) > \mathrm{error}(\mathbf{W}'_f) > \mathrm{error}(\mathbf{W}^*_f)$, holds uniformly across all layers of ResNet18 as well as FFN layers of ViT-B/32. This indicates that the theoretical inequalities are not only valid in principle but also manifest strongly and consistently in real trained models.

Across all layers of ResNet18 and ViT-B/32, the gain from increasing the retained rank by one is consistently negligible, while the gain from replacing pruning with folding is one to two orders of magnitude larger (Fig. 23). Thus, the empirical advantage of folding is not a byproduct of the $k \mapsto k+1$ rank slack, but stems from the richer family of cluster-based projections that folding can realize. This is further corroborated by the layer-wise reconstruction errors in Fig. 24, where $\mathrm{error}(\mathbf{W}_p) > \mathrm{error}(\mathbf{W}'_f) > \mathrm{error}(\mathbf{W}^*_f)$ holds uniformly. These results confirm that folding's superior approximation properties are structural rather than an artifact of rank.

The theory controls the loss via the parameter-Lipschitz bound $|L(\mathbf{W}) - L(\mathbf{W}_\bullet)| \leq \kappa\|\mathbf{W} - \mathbf{W}_\bullet\|_F$. In regimes where $\kappa$ is moderate, *e.g.*, flat solutions obtained with smaller learning rates or SAM-folding's smaller Frobenius error reliably translates into smaller loss degradation. However, in sharp minima such as those produced by Adam at large learning rates (see Fig. 21 and Fig. 22), the effective local $\kappa$ becomes extremely large. In this setting, even tiny parameter perturbations cause large loss changes, and the ordering of Frobenius errors no longer predicts the ordering of accuracies. Thus, the discovered failure cases of folding are not contradictions of the theory but instances where the Lipschitz assumption required for loss control breaks down due to extreme curvature.

# G  USE OF LARGE LANGUAGE MODELS

We used ChatGPT [9] for sentence-level grammar correction and improvement, drafting trivial plotting snippets to produce figures from logs, and code readability edits. All ideas, proofs, experiments, and analyses are ours.

---

[9] ChatGPT / GPT-5: `https://chatgpt.com`

| Layer | Params_fold | Params_mag | $\Delta p$ | FLOPs_fold | FLOPs_mag | $\Delta F$ | Act_fold | Act_mag | $\Delta a$ | NZ_fold | NZ_mag | $\Delta nz$ |
|---|---|---|---|---|---|---|---|---|---|---|---|---|
| classification_head | 513000 | 513000 | 0 | 0 | 0 | 0 | 0 | 0 | 0 | 0 | 0 | 0 |
| transformer.resblocks.0.attn.out_proj | 262656 | 262656 | 0 | 0 | 0 | 0 | 0 | 0 | 0 | 0 | 0 | 0 |
| transformer.resblocks.0.mlp.c_fc | 1050624 | 1050624 | 0 | 0 | 0 | 0 | 0 | 0 | 0 | 0 | 0 | 0 |
| transformer.resblocks.0.mlp.c_proj | 1049088 | 1049088 | 0 | 0 | 0 | 0 | 0 | 0 | 0 | 0 | 0 | 0 |
| transformer.resblocks.1.attn.out_proj | 262656 | 262656 | 0 | 0 | 0 | 0 | 0 | 0 | 0 | 0 | 0 | 0 |
| transformer.resblocks.1.mlp.c_fc | 1050624 | 1050624 | 0 | 0 | 0 | 0 | 0 | 0 | 0 | 0 | 0 | 0 |
| transformer.resblocks.1.mlp.c_proj | 1049088 | 1049088 | 0 | 0 | 0 | 0 | 0 | 0 | 0 | 0 | 0 | 0 |
| transformer.resblocks.10.attn.out_proj | 262656 | 262656 | 0 | 0 | 0 | 0 | 0 | 0 | 0 | 0 | 0 | 0 |
| transformer.resblocks.10.mlp.c_fc | 1050624 | 1050624 | 0 | 0 | 0 | 0 | 0 | 0 | 0 | 0 | 0 | 0 |
| transformer.resblocks.10.mlp.c_proj | 1049088 | 1049088 | 0 | 0 | 0 | 0 | 0 | 0 | 0 | 0 | 0 | 0 |
| transformer.resblocks.11.attn.out_proj | 262656 | 262656 | 0 | 0 | 0 | 0 | 0 | 0 | 0 | 0 | 0 | 0 |
| transformer.resblocks.11.mlp.c_fc | 1050624 | 1050624 | 0 | 0 | 0 | 0 | 0 | 0 | 0 | 0 | 0 | 0 |
| transformer.resblocks.11.mlp.c_proj | 1049088 | 1049088 | 0 | 0 | 0 | 0 | 0 | 0 | 0 | 0 | 0 | 0 |
| transformer.resblocks.2.attn.out_proj | 262656 | 262656 | 0 | 0 | 0 | 0 | 0 | 0 | 0 | 0 | 0 | 0 |
| transformer.resblocks.2.mlp.c_fc | 1050624 | 1050624 | 0 | 0 | 0 | 0 | 0 | 0 | 0 | 0 | 0 | 0 |
| transformer.resblocks.2.mlp.c_proj | 1049088 | 1049088 | 0 | 0 | 0 | 0 | 0 | 0 | 0 | 0 | 0 | 0 |
| transformer.resblocks.3.attn.out_proj | 262656 | 262656 | 0 | 0 | 0 | 0 | 0 | 0 | 0 | 0 | 0 | 0 |
| transformer.resblocks.3.mlp.c_fc | 1050624 | 1050624 | 0 | 0 | 0 | 0 | 0 | 0 | 0 | 0 | 0 | 0 |
| transformer.resblocks.3.mlp.c_proj | 1049088 | 1049088 | 0 | 0 | 0 | 0 | 0 | 0 | 0 | 0 | 0 | 0 |
| transformer.resblocks.4.attn.out_proj | 262656 | 262656 | 0 | 0 | 0 | 0 | 0 | 0 | 0 | 0 | 0 | 0 |
| transformer.resblocks.4.mlp.c_fc | 1050624 | 1050624 | 0 | 0 | 0 | 0 | 0 | 0 | 0 | 0 | 0 | 0 |
| transformer.resblocks.4.mlp.c_proj | 1049088 | 1049088 | 0 | 0 | 0 | 0 | 0 | 0 | 0 | 0 | 0 | 0 |
| transformer.resblocks.5.attn.out_proj | 262656 | 262656 | 0 | 0 | 0 | 0 | 0 | 0 | 0 | 0 | 0 | 0 |
| transformer.resblocks.5.mlp.c_fc | 1050624 | 1050624 | 0 | 0 | 0 | 0 | 0 | 0 | 0 | 0 | 0 | 0 |
| transformer.resblocks.5.mlp.c_proj | 1049088 | 1049088 | 0 | 0 | 0 | 0 | 0 | 0 | 0 | 0 | 0 | 0 |
| transformer.resblocks.6.attn.out_proj | 262656 | 262656 | 0 | 0 | 0 | 0 | 0 | 0 | 0 | 0 | 0 | 0 |
| transformer.resblocks.6.mlp.c_fc | 1050624 | 1050624 | 0 | 0 | 0 | 0 | 0 | 0 | 0 | 0 | 0 | 0 |
| transformer.resblocks.6.mlp.c_proj | 1049088 | 1049088 | 0 | 0 | 0 | 0 | 0 | 0 | 0 | 0 | 0 | 0 |
| transformer.resblocks.7.attn.out_proj | 262656 | 262656 | 0 | 0 | 0 | 0 | 0 | 0 | 0 | 0 | 0 | 0 |
| transformer.resblocks.7.mlp.c_fc | 1050624 | 1050624 | 0 | 0 | 0 | 0 | 0 | 0 | 0 | 0 | 0 | 0 |
| transformer.resblocks.7.mlp.c_proj | 1049088 | 1049088 | 0 | 0 | 0 | 0 | 0 | 0 | 0 | 0 | 0 | 0 |
| transformer.resblocks.8.attn.out_proj | 262656 | 262656 | 0 | 0 | 0 | 0 | 0 | 0 | 0 | 0 | 0 | 0 |
| transformer.resblocks.8.mlp.c_fc | 1050624 | 1050624 | 0 | 0 | 0 | 0 | 0 | 0 | 0 | 0 | 0 | 0 |
| transformer.resblocks.8.mlp.c_proj | 1049088 | 1049088 | 0 | 0 | 0 | 0 | 0 | 0 | 0 | 0 | 0 | 0 |
| transformer.resblocks.9.attn.out_proj | 262656 | 262656 | 0 | 0 | 0 | 0 | 0 | 0 | 0 | 0 | 0 | 0 |
| transformer.resblocks.9.mlp.c_fc | 1050624 | 1050624 | 0 | 0 | 0 | 0 | 0 | 0 | 0 | 0 | 0 | 0 |
| transformer.resblocks.9.mlp.c_proj | 1049088 | 1049088 | 0 | 0 | 0 | 0 | 0 | 0 | 0 | 0 | 0 | 0 |
| visual.conv1 | 2359296 | 2359296 | 0 | 115605504 | 115605504 | 0 | 37632 | 37632 | 0 | 37632 | 37632 | 0 |
| visual.transformer.resblocks.0.attn.out_proj | 590592 | 590592 | 0 | 0 | 0 | 0 | 0 | 0 | 0 | 0 | 0 | 0 |
| visual.transformer.resblocks.0.mlp.c_fc | 1889433 | 1889433 | 0 | 94348800 | 94348800 | 0 | 122850 | 122850 | 0 | 122850 | 122850 | 0 |
| visual.transformer.resblocks.0.mlp.c_proj | 1887744 | 1887744 | 0 | 94348800 | 94348800 | 0 | 38400 | 38400 | 0 | 38400 | 38400 | 0 |
| visual.transformer.resblocks.1.attn.out_proj | 590592 | 590592 | 0 | 0 | 0 | 0 | 0 | 0 | 0 | 0 | 0 | 0 |
| visual.transformer.resblocks.1.mlp.c_fc | 1889433 | 1889433 | 0 | 94348800 | 94348800 | 0 | 122850 | 122850 | 0 | 122850 | 122850 | 0 |
| visual.transformer.resblocks.1.mlp.c_proj | 1887744 | 1887744 | 0 | 94348800 | 94348800 | 0 | 38400 | 38400 | 0 | 38400 | 38400 | 0 |
| visual.transformer.resblocks.10.attn.out_proj | 590592 | 590592 | 0 | 0 | 0 | 0 | 0 | 0 | 0 | 0 | 0 | 0 |
| visual.transformer.resblocks.10.mlp.c_fc | 1889433 | 1889433 | 0 | 94348800 | 94348800 | 0 | 122850 | 122850 | 0 | 122850 | 122850 | 0 |
| visual.transformer.resblocks.10.mlp.c_proj | 1887744 | 1887744 | 0 | 94348800 | 94348800 | 0 | 38400 | 38400 | 0 | 38400 | 38400 | 0 |
| visual.transformer.resblocks.11.attn.out_proj | 590592 | 590592 | 0 | 0 | 0 | 0 | 0 | 0 | 0 | 0 | 0 | 0 |
| visual.transformer.resblocks.11.mlp.c_fc | 1889433 | 1889433 | 0 | 94348800 | 94348800 | 0 | 122850 | 122850 | 0 | 122850 | 122850 | 0 |
| visual.transformer.resblocks.11.mlp.c_proj | 1887744 | 1887744 | 0 | 94348800 | 94348800 | 0 | 38400 | 38400 | 0 | 38400 | 38400 | 0 |
| visual.transformer.resblocks.2.attn.out_proj | 590592 | 590592 | 0 | 0 | 0 | 0 | 0 | 0 | 0 | 0 | 0 | 0 |
| visual.transformer.resblocks.2.mlp.c_fc | 1889433 | 1889433 | 0 | 94348800 | 94348800 | 0 | 122850 | 122850 | 0 | 122850 | 122850 | 0 |
| visual.transformer.resblocks.2.mlp.c_proj | 1887744 | 1887744 | 0 | 94348800 | 94348800 | 0 | 38400 | 38400 | 0 | 38400 | 38400 | 0 |
| visual.transformer.resblocks.3.attn.out_proj | 590592 | 590592 | 0 | 0 | 0 | 0 | 0 | 0 | 0 | 0 | 0 | 0 |
| visual.transformer.resblocks.3.mlp.c_fc | 1889433 | 1889433 | 0 | 94348800 | 94348800 | 0 | 122850 | 122850 | 0 | 122850 | 122850 | 0 |
| visual.transformer.resblocks.3.mlp.c_proj | 1887744 | 1887744 | 0 | 94348800 | 94348800 | 0 | 38400 | 38400 | 0 | 38400 | 38400 | 0 |
| visual.transformer.resblocks.4.attn.out_proj | 590592 | 590592 | 0 | 0 | 0 | 0 | 0 | 0 | 0 | 0 | 0 | 0 |
| visual.transformer.resblocks.4.mlp.c_fc | 1889433 | 1889433 | 0 | 94348800 | 94348800 | 0 | 122850 | 122850 | 0 | 122850 | 122850 | 0 |
| visual.transformer.resblocks.4.mlp.c_proj | 1887744 | 1887744 | 0 | 94348800 | 94348800 | 0 | 38400 | 38400 | 0 | 38400 | 38400 | 0 |
| visual.transformer.resblocks.5.attn.out_proj | 590592 | 590592 | 0 | 0 | 0 | 0 | 0 | 0 | 0 | 0 | 0 | 0 |
| visual.transformer.resblocks.5.mlp.c_fc | 1889433 | 1889433 | 0 | 94348800 | 94348800 | 0 | 122850 | 122850 | 0 | 122850 | 122850 | 0 |
| visual.transformer.resblocks.5.mlp.c_proj | 1887744 | 1887744 | 0 | 94348800 | 94348800 | 0 | 38400 | 38400 | 0 | 38400 | 38400 | 0 |
| visual.transformer.resblocks.6.attn.out_proj | 590592 | 590592 | 0 | 0 | 0 | 0 | 0 | 0 | 0 | 0 | 0 | 0 |
| visual.transformer.resblocks.6.mlp.c_fc | 1889433 | 1889433 | 0 | 94348800 | 94348800 | 0 | 122850 | 122850 | 0 | 122850 | 122850 | 0 |
| visual.transformer.resblocks.6.mlp.c_proj | 1887744 | 1887744 | 0 | 94348800 | 94348800 | 0 | 38400 | 38400 | 0 | 38400 | 38400 | 0 |
| visual.transformer.resblocks.7.attn.out_proj | 590592 | 590592 | 0 | 0 | 0 | 0 | 0 | 0 | 0 | 0 | 0 | 0 |
| visual.transformer.resblocks.7.mlp.c_fc | 1889433 | 1889433 | 0 | 94348800 | 94348800 | 0 | 122850 | 122850 | 0 | 122850 | 122850 | 0 |
| visual.transformer.resblocks.7.mlp.c_proj | 1887744 | 1887744 | 0 | 94348800 | 94348800 | 0 | 38400 | 38400 | 0 | 38400 | 38400 | 0 |
| visual.transformer.resblocks.8.attn.out_proj | 590592 | 590592 | 0 | 0 | 0 | 0 | 0 | 0 | 0 | 0 | 0 | 0 |
| visual.transformer.resblocks.8.mlp.c_fc | 1889433 | 1889433 | 0 | 94348800 | 94348800 | 0 | 122850 | 122850 | 0 | 122850 | 122850 | 0 |
| visual.transformer.resblocks.8.mlp.c_proj | 1887744 | 1887744 | 0 | 94348800 | 94348800 | 0 | 38400 | 38400 | 0 | 38400 | 38400 | 0 |
| visual.transformer.resblocks.9.attn.out_proj | 590592 | 590592 | 0 | 0 | 0 | 0 | 0 | 0 | 0 | 0 | 0 | 0 |
| visual.transformer.resblocks.9.mlp.c_fc | 1889433 | 1889433 | 0 | 94348800 | 94348800 | 0 | 122850 | 122850 | 0 | 122850 | 122850 | 0 |
| visual.transformer.resblocks.9.mlp.c_proj | 1887744 | 1887744 | 0 | 94348800 | 94348800 | 0 | 38400 | 38400 | 0 | 38400 | 38400 | 0 |
| TOTALS | 83633940 | 83633940 | 0 | 2379976704 | 2379976704 | 0 | 1972632 | 1972632 | 0 | 1972632 | 1972632 | 0 |

Table 10: Per-layer comparison of CLIP ViT-B/32 after `FOLD` and `MAG2` at compression ratio 0.2 (global parameter reduction 7.47%). The table reports all convolutional and linear layers in the vision transformer and classification head, including their parameters, per-image FLOPs, activation sizes, and effective activations. The `transformer.resblocks.*` modules belong to CLIP's text encoder. Because the ImageNet-1k fine-tuned variant evaluates only the vision encoder and classification head, the text encoder is not part of the forward graph. THOP therefore records zero FLOPs and zero activations for these layers, while their parameters remain included in the model. Note that the per-layer totals ($\approx 8.36 \times 10^7$ parameters) are smaller than the full model parameter count ($\approx 1.40 \times 10^8$) because this table excludes components without FLOPs, such as token embeddings, positional embeddings, and LayerNorm parameters, which are included in the global counts but not part of the per-layer FLOP/activation analysis. Several projection layers inside the attention blocks show zero FLOPs because CLIP implements attention using fused operations; these operations are profiled at the block level by THOP rather than attributed to the individual Linear submodules. All per-layer differences are zero, showing that `FOLD` and `MAG2` produce structurally identical compressed models on every layer affected by compression.

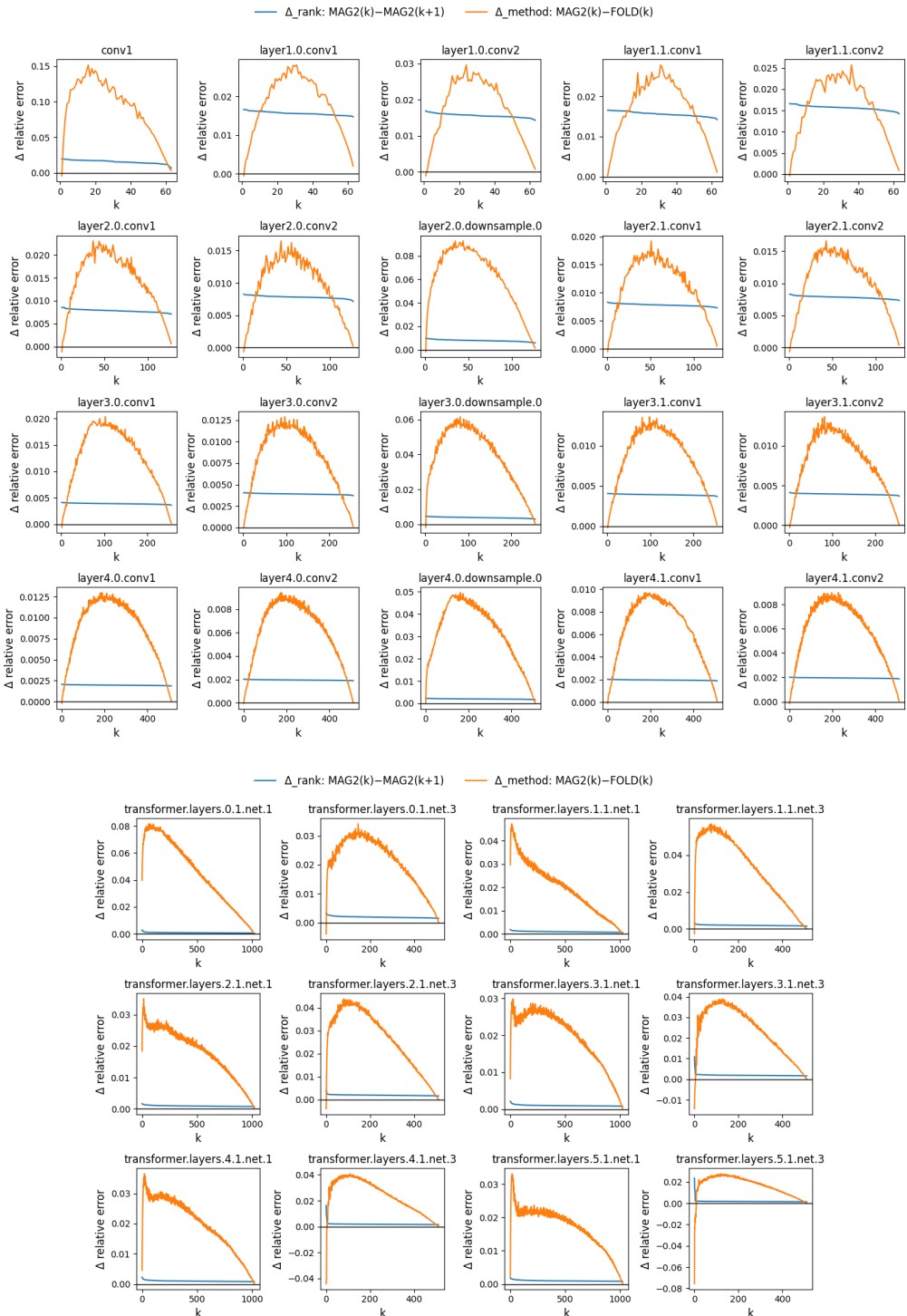

Figure 23: **Effect of increasing the retained rank by one is significantly lower than changing the compression method from MAG to FOLD.** Comparison of (i) the effect of increasing the retained rank by one (blue curves) and (ii) the effect of switching from MAG2 to FOLD at the same nominal rank (orange curves). Each panel corresponds to a single weight matrix **W** in ResNet18 convolutional layers (**top**) and ViT-B/32 FFN layers (**bottom**) and shows the relative Frobenius error difference $\Delta$ as a function of the retained rank $k$.

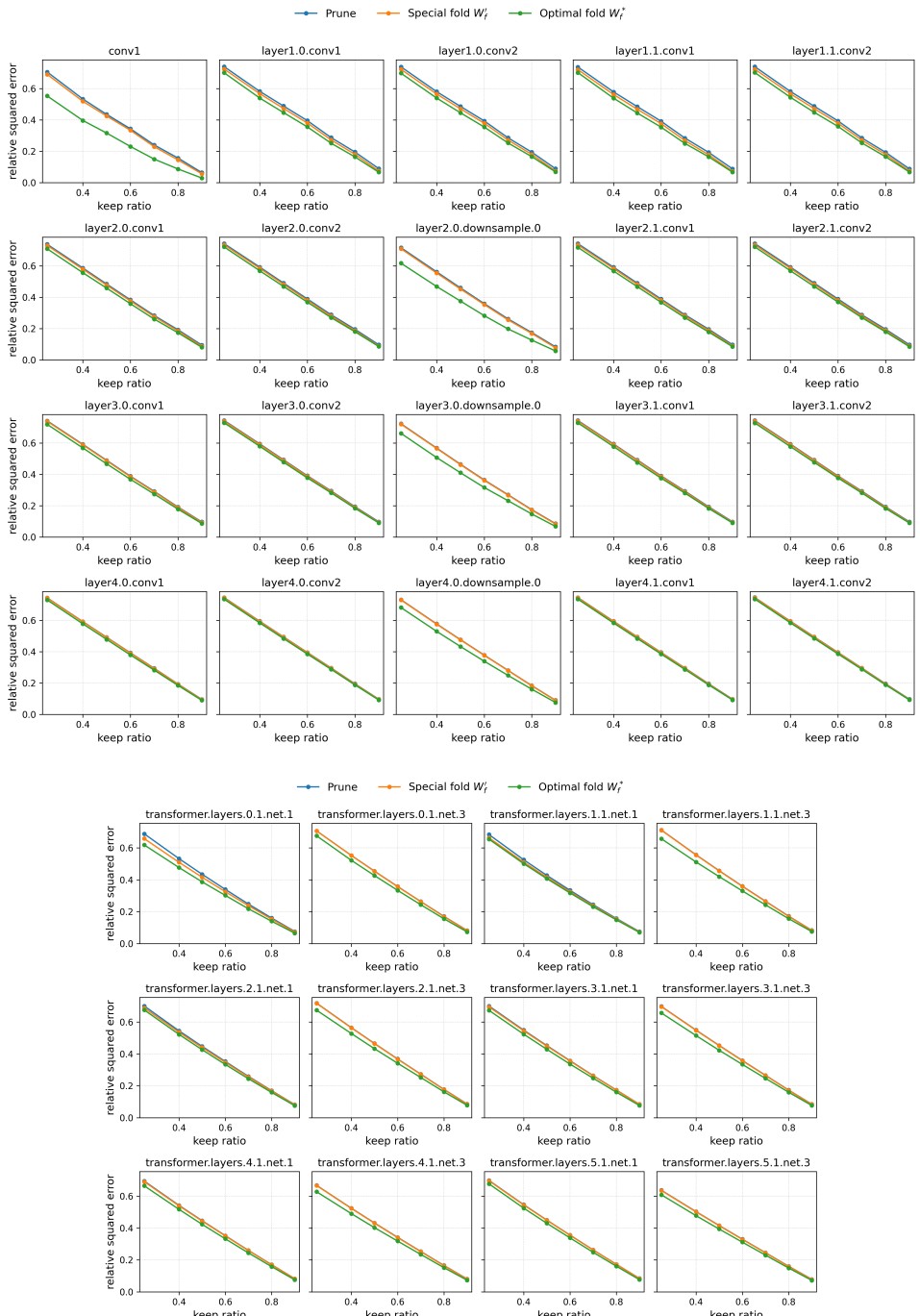

Figure 24: **Relative reconstruction error of pruning, special folding $\mathbf{W}'_f$ (from the proof of Theorem 2.1), and optimal folding $\mathbf{W}^*_f$ for all layers of ResNet18 (top) and ViT-B/32 (bottom).** For each layer, we report the normalized squared Frobenius error $\|\mathbf{W} - \mathbf{W}_\bullet\|^2_F / \|\mathbf{W}\|^2_F$ at several keep ratios $k_p/m$, where $\bullet \in \{\texttt{MAG2}, \texttt{singleton}, \texttt{FOLD}\}$. MAG2 (blue) denotes structured magnitude pruning with $k_p$ retained rows. The special fold $\mathbf{W}'_f$ (orange) merges all pruned rows into a single extra cluster ($k_f = k_p + 1$). The optimal fold $\mathbf{W}^*_f$ (green) is the $k$-means solution with $k_f$ clusters. Across all layers, $\mathbf{W}'_f$ consistently outperforms pruning, and $\mathbf{W}^*_f$ yields the smallest error, empirically validating $\text{error}(\mathbf{W}_p) \geq \text{error}(\mathbf{W}'_f) \geq \text{error}(\mathbf{W}^*_f)$.

