# OpenReview forum: "Cut Less, Fold More: Model Compression through the Lens of Projection Geometry"
_ICLR.cc/2026/Conference — ICLR 2026 Poster_

### Official Review · Reviewer_kdrS · 2025-10-31

**Soundness:** 2
**Presentation:** 4
**Contribution:** 2
**Rating:** 2
**Confidence:** 4

**Summary:**

The paper builds on model folding introduced in Wang et al. (2025), comparing it to structured magnitude pruning both theoretically and empirically. The authors show that both structured pruning and folding can be viewed as orthogonal projections in parameter space, and based on that show that folding has leads to a smaller reconstruction error with one-rank slack. Experiments on more than 1,000 checkpoints of CNNs and ViTs show that folding leads to a higher post-compression accuracy than structured magnitude pruning in most cases, especially at moderate to high compression; even without the one-rank slack assumed for the theoretical results. In-depth ablation studies show that the improvement of folding over structured pruning are most pronounced in training settings that promote convergence to flatter local minima.

**Strengths:**

Overall, the paper is written well, easy to follow, and the reader is not left with many questions. The experiments are executed well, showing results for a multitude of models over four architectures, and in-depth ablation studies are provided. The experiments without post-compression reconstruction show that in most cases, FOLD outperforms MAG, with the gaps increasing with compression ratio, validating the findings in Wang et al. (2025). It is interesting to see how the training procedure impacts the performance of the two post-training compression techniques studied in the paper. Experiments comparing FOLD and MAG with post compression fine tuning of only normalization layers or the full models provide novel insights on how folding and magnitude pruning compare in this setting.

**Weaknesses:**

I have several concerns regarding soundness and contribution of the work, which I explain below. I hope that these remarks are helpful in improving the work and I am happy to discuss my evaluation.

- Theorems 2.1 and 2.2 compare structured magnitude pruning with $k$ retained rows and folding with $k+1$ retained rows and conclude that folding, in theory, outperforms pruning. However, the error inequalities given in the theorems say nothing about the performance of folding compared to pruning as pruning with $k+1$ retained rows also outperforms pruning with $k$ retained rows.
- The claim of "folding incurs provably smaller functional deviation" seems to refer to the output or decision of the model, but is based on the parameter-Lipschitz assumption of the loss. This bound does not control the change in model output directly.
- The setup is called "calibration-free" yet for CNNs, there is no evaluation without data-based post-pruning reconstruction. Even in the no fine-tuning setting, REPAIR is used to re-estimate the batch-normalization.
- The experiments are conducted only with small scale architectures, the largest being CLIP ViT-B-32 at 151M parameters. The authors point out that calibration-free structured pruning methods collapse in the LLM pruning setting. However, experiments on larger vision transformers like ViT-22B would be interesting.
- The figures in the paper mostly contain plots comparing model quality after FOLD and MAG compression but not the quality of the uncompressed models. In some cases, this makes it hard to judge if low accuracy is due to compression performing badly or the base model itself having low accuracy - it might be the case that FOLD performs better especially in the cases where the base model itself is weak.
- Minor typo: In line 146, the loss difference is measured using the Frobenius norm.

**Questions:**

- How does the pre-compression model accuracy relate to delta accuracy? This is especially interesting since augmentation is both known to strongly impact final model performance and has a large impact on delta accuracy.

---

> ### Author Response · Authors · 2025-11-24
> **Response to Reviewer kdrS (1/2)**
>
> We thank the reviewer for the thorough and constructive feedback. We hope to have successfully clarified the reviewer’s concerns below.
>
> **Theoretical claims.**
> We suspect there is a small typo in the reviewer’s comment ("pruning with  retained rows also outperforms pruning with retained rows"), and our best interpretation is that the concern refers to the fact that Theorems 2.1-2.2 analyze pruning and folding as projections on the weight matrix W without explicitly removing output rows and adjusting the downstream layer. If read this way, the reviewer is right that the theoretical comparison is made at the level of the pre-activation Wx: we study the functional deviation at the input of the nonlinearity, where a Lipschitz condition ensures that smaller perturbations imply smaller local loss changes. A more “downstream” comparison (after the nonlinearity and next weight matrix) would mix in the structure of the subsequent layer and no longer isolate the projection operator itself. Importantly, removing rows of W after projection does not change the network’s function once the next-layer weights are adjusted accordingly. This is exactly what we do in our implementation, and all experiments enforce strict parameter/FLOP equality (see Tab. 7-10).
>
> **Theoretical bound and model decision boundary.**
> We thank the reviewer for highlighting this point. Our statement that “folding incurs provably smaller functional deviation” is meant strictly in terms of loss perturbation, not in terms of pointwise output or logits. Theorem 2.2 uses a parameter-Lipschitz assumption on the loss as a local smoothness condition, which is the standard analytic tool in post-training compression: it provides a sufficient (though not necessary) guarantee that smaller parameter perturbations imply smaller loss deviations in a neighborhood of the pretrained solution. We agree that this does not directly bound the change in the output function, and we clarify this in the paper (see our changes throughout the theoretical part).
>
> **Evaluation before REPAIR.**
> For CNNs, we now include strictly calibration-free results before REPAIR in Fig. 17\. These reflect the raw effect of pruning/folding. However, structured pruning of BatchNorm networks is known to cause variance/signal collapse, as documented in the original folding paper \[1\], REPAIR paper \[2\] and in recent work on pruning \[3\]. Because this collapse is unrelated to the pruning/folding mechanism itself, prior work treats a lightweight BN-statistic correction as part of the standard pipeline. Following this practice, we apply REPAIR using a single unlabeled forward pass (no labels, no gradients, no optimization) to both methods.
>
> For ViTs, no BatchNorm is present, so post-compression results (Fig. 1\) already represent the true data-free behavior.
>
> \[1\] D. Wang et. al., Forget the Data and Fine-Tuning\! Just Fold the Network to Compress, [https://arxiv.org/abs/2502.10216](https://arxiv.org/abs/2502.10216)
>
> \[2\] K. Jordan, REPAIR: REnormalizing Permuted Activations for Interpolation Repair, [https://arxiv.org/abs/2211.08403](https://arxiv.org/abs/2211.08403)
>
> \[3\] D. Saikumar, Signal Collapse in One-Shot Pruning: When Sparse Models Fail to Distinguish Neural Representations, [https://arxiv.org/abs/2502.15790](https://arxiv.org/abs/2502.15790)

---

> ### Author Response · Authors · 2025-11-24
> **Response to Reviewer kdrS (2/2)**
>
> **Performance on large-scale architecture.**
> We agree that evaluating larger models is important. In the revised manuscript, we now include additional experiments on LLaMA-60M and LLaMA-130M, each trained under several distinct hyperparameter configurations (see Tab. 1 in Sec. 3 and Tab. 6 in Appendix G). These results demonstrate that our geometric observations extend beyond CNNs and ViTs to modern transformer architectures. We also reference the original model folding work \[1\], which reports results on LLaMA-7B, further supporting scalability.
>
> Training and sweeping many large-scale models across diverse hyperparameters, required for our comparison methodology, is computationally expensive. We therefore focus our extended analysis on smaller LLaMA and ViT variants, which still demonstrate that our conclusions hold beyond CIFAR-10/ImageNet-1k and beyond CNNs. These architectures and model sizes are representative of practical, resource-constrained deployments, such as edge and mobile systems, making them an important and relevant class.
>
> We would be very happy to include results on even larger vision transformers such as ViT-22B. However, we were unable to find public checkpoints trained under varying hyperparameters, and training such models ourselves is beyond our compute budget. If the reviewer could point us to (or share) checkpoints for ViT-22B trained under different optimization settings, we will process them and report the results in the final version.
>
> \[1\] D. Wang et. al., Forget the Data and Fine-Tuning\! Just Fold the Network to Compress, [https://arxiv.org/abs/2502.10216](https://arxiv.org/abs/2502.10216)
>
> **The quality of uncompressed models.**
> We added Fig. 18 in Appendix G (see attached PDF), which explicitly plots the accuracy of the uncompressed models against the performance difference between FOLD and MAG2. The results show that FOLD does not perform better on weak models. Instead, its advantage grows with the base model’s accuracy. High-performing models, most notably ResNet18 (SGD) and ViT-B/32, consistently exhibit larger positive values of ΔAccuracy \= Acc(FOLD) − Acc(MAG). Hence, FOLD’s benefit is amplified on strong models.
>
> The reviewer is concerned that data augmentation might bias our results. The checkpoints we trained ourselves as well as those obtained from prior work and used in our study do not exhibit any known imbalance in hyperparameter choices, including data augmentation. All training details are provided in Appendix D. In our ablation studies we filter checkpoints according to the criteria stated in each figure, ensuring that all comparisons are made across models trained under consistent conditions.
>
> **Typo**.
> Thank you for pointing this out. The revised manuscript now fixes this.

---

> > ### Comment · Reviewer_kdrS · 2025-11-26
> >
> > Thank you for the detailed rebuttal.
> >
> > > We suspect there is a small typo in the reviewer’s comment ("pruning with retained rows also outperforms pruning with retained rows")
> >
> > Regarding the theoretical claims, I assume the MathJax in the review did not render correctly, it does for me. My concern is that Theorems 2.1 and 2.2 compare structured magnitude pruning with k retained rows to folding with k+1 retained rows, and then conclude that folding theoretically outperforms pruning. However, the error bounds in these theorems do not substantiate this comparison, as pruning with k+1 retained rows would likewise outperform pruning with k retained rows.
> >
> > The rest of my concerns have been adequately addressed.

---

> ### Author Response · Authors · 2025-12-01
>
> Yes, the misunderstanding was due to math rendering - thank you for the clarification.
>
>
> **Why the result is more than “rank+1 is better than rank”.**
>
> It is of course true that, within a fixed family of projections, increasing the rank from k to k+1 can reduce approximation error (e.g. pruning with k+1 units vs. pruning with k units), as reviewer correctly notes. What Theorems 2.1-2.2 show is stronger:
> - For any pruning projection C_p​ of rank k​, we can construct a specific folding projection C’_f​ of rank k+1 whose error is always smaller.
> - Moreover, this C’_f​ is a suboptimal folding obtained only to link pruning and folding. The optimal k-means folding C*_f​ can only improve this error further:
> ||W−W_p||^2_F >= ||W−W'_f||^2_F >= ​ ||W−W*_f||^2_F.
> As noted in the paper and shown empirically in the new Appendix G.3 (see the detailed explanation below), the improvement stems only negligibly from a rank slack of one. It derives mainly from the broader class of cluster-structured projections that folding enables compared with axis-aligned pruning.
> - Finally, the seemingly symmetric statement is not true: It's not true that pruning rank k+1 is always better than folding rank k (a counterexample is if all rows of W are identical).
>
>
> **Why the rank slack of one is practically negligible.**
>
> In typical settings, layers are wide and we prune many channels. Under a 50% retention, a ResNet-18 stage with 256 channels keeps k=128 (so folding uses k+1=129), and a ViT-B/32 block with width 768 keeps k=384 (so k+1=385). The relative rank increase is 1/k≈0.78% and 0.26%, respectively. In Appendix G.3 (new), we empirically show that:
> - The Frobenius reconstruction error ||W - W_k||_F as a function of rank k is smooth, and the typical change from k to k+1 is negligible (Fig. 23).
> - For fixed keep ratios, pruning always yields the largest reconstruction error, the special folding W'_f consistently improves over pruning, and the optimal folding W*_f achieves the smallest error, empirically validating error(W_p) >= error(W'_f) >= error(W*_f) across layers (Fig. 24).
> - The observed accuracy changes when increasing k by one are substantially smaller than the typical gap between pruning and folding at comparable compression rates. This supports the interpretation that geometry (axis-aligned vs. cluster-structured projections) rather than the extra rank is the main driver of the performance differences we observe.
>
>
> **On equal-rank comparisons and failure cases.**
>
> We do not claim that folding is universally better than pruning at identical rank or for every training setup. Indeed, we observe empirical “failure” cases for folding (e.g., very sharp solutions obtained by Adam with large learning rate), which indicate that parameter-space Frobenius distance alone can be insufficient to control the loss in such regimes. This is consistent with our analysis, which provides only Lipschitz-style upper bounds on the loss perturbation.
>
> At the same time, pruning and folding belong to fundamentally different projection families, and the observed empirical advantage of folding is not merely a trivial consequence of adding one more retained row.
>
>
> **Changes and clarifications in the updated version of the paper.**
>
> In the updated version of our paper, we:
> - revise the notation in the surrounding text of Theorems 2.1-2.2 to highlight the differences between U’_f, C’_f, W’_f ​vs. U*_f,C*_f,W*_f.
> - add empirical plots illustrating how small the effect of increasing rank by one is compared to the gap between pruning and folding (Fig. 23).
> - include visualizations that empirically verify the theoretical chain ||W−W_p||^2_F >= ||W−W'_f||^2_F >= ​ ||W−W*_f||^2_F, showing that these inequalities hold consistently across layers and match the behavior predicted by Theorems 2.1-2.2 (Fig. 24).
> - add a short discussion clarifying the limitations of the Lipschitz-based argument in sharp regions of the loss landscape (Appendix G.3).
>
> We thank the reviewer for the careful examination of our theoretical claims and for the constructive feedback that helped us clarify and improve the paper.

---

### Official Review · Reviewer_y5Wu · 2025-11-02

**Soundness:** 3
**Presentation:** 3
**Contribution:** 3
**Rating:** 6
**Confidence:** 3

**Summary:**

The paper frames structured pruning and model folding as orthogonal projections in parameter space: pruning is an axis-aligned projection, while folding is a low-rank, cluster-structured projection. The authors prove that folding yields smaller parameter reconstruction error and tighter (Lipschitz-based) bounds on functional perturbation than pruning. A large-scale, calibration-free study shows folding usually achieves higher post-compression accuracy, especially at moderate to high compression, and it keeps its edge after lightweight or short fine-tuning.

**Strengths:**

- Casting pruning and folding as orthogonal projections cleanly explains their geometric differences; theorems show folding’s strictly smaller projection error and hence tighter loss perturbation under mild smoothness

- The evaluation spans >1000 checkpoints and multiple architectures/datasets, with consistent wins for folding at moderate to high compression and robustness to training variations.

- Folding’s advantage persists after minimal recalibration (e.g., BN/LayerNorm reset) and 1-5 epochs of fine-tuning, which mirrors realistic deployment where full retraining is expensive.

**Weaknesses:**

- The core guarantee uses a one-rank slack (pruning rank vs. folding rank)  Although the authors state experiments match retained parameters/FLOPs, a theoretical result at exactly matched rank would remove any residual ambiguity, especially at low compression where a single unit can matter. Consider strengthening theory (e.g., conditions under which folding dominates at equal rank) or adding tighter empirical per-layer equality checks to preclude hidden capacity differences.

- Results focus on accuracy; there’s little on real-world latency, memory bandwidth, or kernel fusion performance after folding - despite identical FLOPs at matched size, wall-clock behavior can differ.

**Questions:**

Can you provide per-layer tables verifying that, for every comparison, parameters and FLOPs (and effective activations) are exactly matched between folding and pruning after the next-layer adaptation, not just approximately matched globally? If there are exceptions, quantify their impact.

---

> ### Author Response · Authors · 2025-11-24
> **Response to Reviewer y5Wu**
>
> We thank the reviewer for the insightful comments. The runtime analysis appears in the updated manuscript at the end of Appendix G. All new additions and changes to the paper are marked in blue in the PDF.
>
> **The core guarantee uses a one-rank slack.**
> We thank the reviewer for bringing this up. The one-rank slack in Theorems 2.1-2.2 is a technical device that enables the guarantee to hold for arbitrary pruning masks. Coordinate projections (pruning) can zero out directions that are essential for reconstructing a given weight matrix at a fixed subspace dimension, allowing one additional dimension removes this degenerate corner case and yields a clean, general dominance statement. Importantly, this slack is never used in our experiments: folding and pruning always match exactly in per-layer rank, parameter count, and FLOPs, and Appendix G now includes explicit layer-wise equality checks to eliminate any ambiguity (Tab. 9 and Tab. 10).
>
> We agree that an equal-rank theoretical dominance result would be even stronger. However, without extra assumptions on weight or landscape structure, such a statement cannot hold in general, because coordinate projections can be arbitrarily misaligned with the underlying weight geometry. While our current theory focuses on the most general setting, we have extended the paper with additional empirical analyses of sharpness showing when folding’s advantage persists in practice. Developing theoretical conditions under which equal-rank dominance can be guaranteed is an interesting direction that we are excited to pursue in future work.
>
> **Runtime overhead.**
> We added comprehensive runtime and memory profiling for all methods evaluated in this work. The results for PreActResNet18 (Tab. 7 and Tab. 9, Appendix G) and CLIP ViT-B/32 (Tab. 8 and Tab. 10, Appendix G) report compression time, peak compression memory, inference latency, FLOPs, and forward-pass memory before and after compression. These measurements show that all methods yield identical compressed architectures, and therefore identical FLOPs and inference latency. Folding incurs only a moderate compression overhead (due to the iterative k-means algorithm), which is small relative to the cost of fine-tuning or model evaluation (seconds to a minute vs. minutes to hours).
>
> **Can you provide per-layer tables verifying matched parameters / FLOPS?**
> Yes, please see Table 9 and Table 10 in Appendix G in the PDF. We provide the results for PreActResNet18 and CLIP ViT-B/32 backbones. The respective scripts and their outputs used to produce the tables are in our anonymous repository together with the source code (link in Appendix A).

---

### Official Review · Reviewer_b6Qf · 2025-11-03

**Soundness:** 2
**Presentation:** 2
**Contribution:** 2
**Rating:** 4
**Confidence:** 5

**Summary:**

This paper studies calibration-free model compression by framing pruning and model folding as orthogonal projections in parameter space. The authors argue that folding (weight clustering) preserves functional similarity better than pruning (weight removal) and provide both theoretical guarantees and large-scale empirical validation over 1,000 CNN and ViT checkpoints across CIFAR-10 and ImageNet-1K. Folding is shown to outperform pruning, especially at moderate-to-high compression levels.

**Strengths:**

- The projection-theoretic framing of pruning and folding is reasonable. Framing compression as orthogonal projection provides an solid interpretation.

- The paper is writing well with good visualizations.

**Weaknesses:**

- The main argument is questionable. The paper argues that model folding is better than model pruning with closer distance to the original models. This statement is a little problematic. Since pruning domain has been developed broadly and deeply in the past decade, many pruning methods could produce pruning models even surpassing original models. The logic here is that pruning could eliminate some harmful neurons to make the pruned model to be stronger. Therefore, the distance to original model is not sufficiently strong and convincing enough to support the outperformance.

- Numerical experiment is limited. The paper only compares with basic magnitude pruning schema. Given theoretical support being not sufficient, it should compare and discuss with more recent representative works, e.g., DepGraph, OTO, etc., which contains more pruning protocols.

**Questions:**

See the weaknesses.

---

> ### Author Response · Authors · 2025-11-24
> **Response to Reviewer b6Qf (1/2)**
>
> We thank the reviewer for the thorough and constructive feedback. We hope to have successfully clarified the reviewer’s concerns below.
>
> **The main argument is questionable**
> We agree that post-training compression can sometimes improve accuracy, e.g., by removing low-signal channels or inducing a slightly better effective structure. These effects are real but incidental: they depend on local redundancies in a specific checkpoint. Our theoretical analysis is about operator-level projection quality, not about ruling out such incidental gains. Pruning and folding are formalized as two orthogonal projection operators onto lower-dimensional subspaces of equal capacity: pruning corresponds to an axis-aligned coordinate projection, while folding projects onto a cluster-structured subspace that better reflects the empirical correlation structure of trained weights. Theorems 2.1-2.2 show that, for any pruned solution, there exists a folded solution with (almost) the same rank that achieves strictly smaller reconstruction error in Frobenius norm.
>
> The Lipschitz continuity assumption in Eq. (1) is indeed idealized if read as a global property, and we do not claim that real networks are well-behaved in this sense. In practice, the relevant regime is local: after training, the model sits near a stationary point, and standard local smoothness arguments (as widely used in post-training quantization and distillation) imply that smaller parameter perturbations yield tighter local bounds on loss change. Our use of distance-to-the-original-model is therefore not a universal performance predictor but a principled way to compare projection operators under matched capacity: folding gives a uniformly tighter worst-case bound on loss perturbation around the pretrained solution, whereas the occasional accuracy improvements the reviewer mentions are non-systematic side effects that can occur for both operators and are not the mechanism behind the consistent folding gains we observe.
>
> To further substantiate these claims, we expanded Appendix G.1 with an extensive sharpness analysis across 200/50/72 independently trained ResNet18, PreActResNet18, and CLIP ViT-B/32 models. The results show that the difference in worst-case adaptive ℓ∞ sharpness between folding and pruning is a strong and consistent predictor of their accuracy difference at matched capacity: whenever folding induces a smaller sharpness increase, it also achieves higher accuracy. This relationship holds across optimizers, learning rates, SAM radii, and augmentation settings. These findings reinforce that folding’s advantage is not due to chance removal of bad channels but to, in most cases, following a less disruptive compression path in the loss landscape. All new experiments and analyses appear in blue in the updated manuscript.

---

> ### Author Response · Authors · 2025-11-24
> **Response to Reviewer b6Qf (2/2)**
>
> **Numerical experiments are limited.**
> Regarding comparisons to recent pruning methods such as DepGraph and OTO: These methods operate in a different regime from ours. OTO explicitly trains the network from scratch with sparsity-inducing objectives, and DepGraph, as well as the methods listed in its comparison tables, relies on data to compute neuron/channel importance scores and performs data-dependent model compression. In contrast, our work studies post-training, data-free one-shot compression of a pretrained model. DepGraph is orthogonal to our contribution: we also account for structural dependencies (in both pruning and folding the same way), but our focus is on how weight vectors are aggregated within each group, not on how groups are formed. Magnitude-based structured pruning is therefore the appropriate baseline for our theoretical setting, as it corresponds exactly to the axis-aligned projection operator we analyze and requires no data or retraining. We significantly expanded and reorganized the related work in Appendix F to provide a clearer overview of the compression landscape and to more explicitly position our contributions within it. Extending model folding to make use of calibration data in a meaningful way and comparison to data-dependent model compression methods, as the reviewer suggests, is an interesting direction for future work.
>
> To the best of our knowledge, prior evaluations of compression methods typically consider only a single training setup per architecture, making it difficult to understand when a given method succeeds or fails. In contrast, although we focus on only a small set of compression techniques (only calibration-free ones), we evaluate them across a broad spectrum of training regimes, revealing behaviour that is not visible in standard one-setting comparisons. Our results show that even among these few methods, performance can vary substantially across hyperparameters, often reversing which method is preferable and how large is the performance gap. The contribution of our work lies in measuring and explaining these effects \- both theoretically and empirically \- rather than producing an exhaustive head-to-head comparison of many compression algorithms.

---

### Official Review · Reviewer_VBA4 · 2025-11-04

**Soundness:** 4
**Presentation:** 4
**Contribution:** 3
**Rating:** 6
**Confidence:** 4

**Summary:**

This paper introduces a unified theoretical framework showing that model folding and structured pruning are orthogonal projections onto different subspaces, with folding provably achieving smaller parameter reconstruction error by projecting onto cluster-structured rather than axis-aligned subspaces. The extensive empirical validation (>1,000 models across ResNet18, PreActResNet18, ViT-B/32, CLIP ViT-B/32 on CIFAR-10/ImageNet-1K) confirms folding consistently outperforms magnitude pruning, especially at moderate-to-high compression ratios, with advantages persisting through fine-tuning. While limited to vision models, the work makes a valuable contribution by establishing folding as a principled, geometry-aware alternative to pruning that better preserves network functionality, supported by both rigorous theory and comprehensive ablations identifying when folding excels (moderate learning rates, SAM training) versus when the gap narrows (extreme hyperparameters).

**Strengths:**

• **Solid theoretical contribution**: Formalization of pruning and folding as orthogonal projections with provable guarantees on parameter reconstruction error, providing principled understanding of compression methods

• **Extensive empirical validation with thorough ablations**: Over 1,000 checkpoints across diverse architectures (CNNs/ViTs) and datasets, with systematic investigation of learning rates, SAM, augmentation, and regularization effects that clearly identify when folding excels

• **Clear and informative figures**: The scatter plots with color-coded compression ratios and bar plots showing accuracy gaps effectively communicate results across all tested architectures

• **Good organization and presentation**: Paper flows logically from theory to experiments to ablations, with comprehensive appendices providing complete experimental details

**Weaknesses:**

**Major Weaknesses**

- **Limited Scope to Small-Scale Models and Datasets**: Evaluations are confined to relatively small architectures such as ResNet18 (11M parameters) and ViT-B/32 (~86M parameters) on CIFAR-10 and ImageNet-1K. The absence of experiments on large-scale models like LLMs (e.g., GPT series) or diffusion models limits confidence in the method’s scalability and its relevance to modern deployment settings with billions of parameters.

- **Insufficient Comparisons to State-of-the-Art Methods**: Although the paper benchmarks against basic magnitude pruning (L1/L2), it lacks head-to-head comparisons with stronger recent baselines such as SparseGPT [1] and Wanda [2] approaches, despite referencing them in Appendix F. As a result, it remains unclear whether folding offers advantages over cutting-edge compression methods.

- **Lack of Empirical Validation for Some of the Claims**: The paper offers intuitive rationales for hyperparameter effects (e.g., SAM for flatter minima, augmentations for invariant solutions) but lacks direct measurements like Hessian eigenvalues or sharpness metrics to quantify them. It cites literature on optimizers and learning rates influencing curvature but conducts no targeted experiments to verify causal links, leaving claims as unverified hypotheses.

- **Lack of Predictive Metrics**: The paper's ablation studies spot cases where pruning beats folding (e.g., very high learning rates with Adam or large SAM radii), but it fails to create predictive tools like sharpness thresholds, weight sparsity ratios, or loss curvature metrics—to foresee these without tons of tests. This after-the-fact analysis offers little real-world advice on choosing methods.

- **Calibration-Free Claim**: While described as calibration-free, the approach still depends on REPAIR (BatchNorm reset) for CNNs, LayerNorm resets for ViTs, and short fine-tuning (up to 5 epochs) to reach its best results. These requirements imply partial data access and tuning, which conflict with the notion of a truly data-free or zero-shot compression pipeline.

**Minor Weaknesses**

- **Deferred Related Work**: A detailed discussion of prior art is confined to Appendix F due to space constraints, which may force readers to cross-reference for context on novelty, potentially disrupting flow in the main text.

- **Lack of Runtime Analysis**: Folding involves k-means clustering per layer, which is computationally heavier than magnitude-based pruning's simple sorting, but the paper provides no measurements or discussions of overhead, overlooking efficiency trade-offs.

[1] Frantar, E. "SparseGPT: Massive Language Models Can Be Accurately Pruned in One-Shot." arXiv preprint arXiv:2301.00774, January 2023.

[2] Sun, M. "A Simple and Effective Pruning Approach for Large Language Models." arXiv preprint arXiv:2306.11695, June 2023 (published at ICLR 2024).

**Questions:**

see the weaknesses.

---

> ### Author Response · Authors · 2025-11-24
> **Response to Reviewer VBA4 (1/2)**
>
> We thank the reviewer for the insightful comments, which helped to improve our work. Below we address all concerns point-by-point. All new experiments and analyses appear in blue in the updated manuscript (see Sec. 3, Sec. 4, Appendix G).
>
> **Limited Scope to Small-Scale Models and Datasets.**
> We agree that evaluating larger models is important. The revised manuscript now includes experiments on LLaMA-60M and LLaMA-130M, each trained with several distinct hyperparameter configurations, as shown in Sec. 3 and Appendix G (Tab. 1 and Tab. 6). These experiments demonstrate that our geometric observations extend beyond CNNs and ViTs to modern transformer architectures. We additionally reference the original Model Folding work, which already reports results on LLaMA-7B, further supporting scalability. Training many large-scale models across diverse hyperparameters is computationally prohibitive for us, which is why our extended analysis focuses on small LLaMA variants. Nevertheless, the new results illustrate that our conclusions generalize beyond CIFAR-10 and ImageNet-1K.
>
> **Insufficient Comparisons to State-of-the-Art Methods.**
> SparseGPT and Wanda are data-dependent pruning methods that require calibration data or activation statistics. Our paper focuses specifically on data-free, calibration-free, structured, post-training compression. A direct comparison between such fundamentally different settings would not be meaningful or fair, and we now make this more explicit in Sec. 5 and in the overview of related work in Appendix F. We discuss conceptual differences with these approaches, and we point out that designing a data-aware folding method would be a valuable direction for future work, where such comparisons would become appropriate.
>
> **Revised Related Work.**
> We significantly expanded and reorganized the related work in Appendix F to provide a clearer overview of the compression landscape and to more explicitly position our contributions within it. To avoid disrupting the flow in the main text, we also added a brief clarification in the introduction, making the novelty of our approach explicit without requiring a reader to consult Appendix F.
>
> **Lack of Runtime Analysis.**
> We added comprehensive runtime and memory profiling for all methods evaluated in this work. The results for PreActResNet18 (Tab. 7 and Tab. 9, Appendix G) and CLIP ViT-B/32 (Tab. 8 and Tab. 10, Appendix G) report compression time, peak compression memory, inference latency, FLOPs, and forward-pass memory before and after compression. These measurements show that all methods yield identical compressed architectures, and therefore identical FLOPs and inference latency. Folding incurs only a moderate compression overhead (due to the iterative k-means algorithm), which is small relative to the cost of fine-tuning or model evaluation (seconds to a few minutes vs. minutes to hours).

---

> ### Author Response · Authors · 2025-11-24
> **Response to Reviewer VBA4 (2/2)**
>
> **Lack of Empirical Validation for Some of the Claims and Lack of Predictive Metrics.**
> We thank the reviewer for the insightful comments. We add sharpness analysis of predictive indicators in Appendix G.1 (Figs. 19-22). These additions directly address both concerns raised by the reviewer: they provide explicit empirical validation of the curvature-related hypotheses discussed in the paper, and they function as predictive tools that help determine when folding or pruning will be preferable. Specifically, we now provide:
>
> – (i) worst-case adaptive $\\ell\_\\infty$ sharpness curves across architectures and perturbation radii. These results show that although all compression methods move models toward sharper regions, folded models often remain in flatter parts of the landscape than magnitude pruned ones, which in turn correlates with their ability to maintain higher accuracy at higher sparsities.
> – (ii) Pearson/Spearman correlations between the difference in sharpness (Δ-sharpness between FOLD and MAG) and the corresponding accuracy difference (Δ-sharpness) for the same underlying model, computed over 200/50/72 independently trained ResNet18/PreActResNet/CLIP checkpoints. These results show a strong negative correlation as long as the compressed models remain in the same basin. The relationship is pronounced for CNNs and somewhat weaker for CLIP.
> – (iii) an analysis of the sharpness-accuracy relationship conditioned on training hyperparameters (Figs. 21-22). These scatter plots show that the predictive link between Δ-sharpness and Δ-accuracy is not only strong overall but also systematically structured by the optimizer used during training. For Adam (Fig. 21), the correlation is consistently negative across all hyperparameter settings: whenever FOLD increases sharpness more than MAG, it also loses accuracy. At higher learning rates, this effect becomes visible because folding tends to push models into sharper regions of the landscape, while magnitude pruning moves them along a comparatively flatter trajectory. However, the sharpness-accuracy relationship itself remains unchanged. For SGD (Fig. 22), the relationship is still negative but somewhat more dispersed, reflecting that SGD generally converges to flatter regions. SAM and data augmentation further modulate these patterns: increasing the SAM radius reduces variability in Δ-sharpness, while RandAug primarily suppresses outliers without altering the underlying trend. These results confirm that sharpness provides a robust indicator of when folding or pruning will perform better.
>
> The link between model compression performance and sharpness aligns with recent works connecting compression and landscape geometry: AdaSAP \[1\] treats pruning as a sharpness-aware process, and Zhang et al. \[2\] show that feasible pruning ratios depend on intrinsic flatness. Our results support this view \- compression initially increases sharpness while the model remains in its original basin, but sufficiently strong compression forces a transition into a flatter, low-curvature region. Folding follows this trajectory more smoothly, preserving basin structure and yielding lower barriers than magnitude pruning in many cases.
>
> \[1\] A. Bair et. al., Adaptive Sharpness-Aware Pruning for Robust Sparse Networks, [https://arxiv.org/abs/2306.14306](https://arxiv.org/abs/2306.14306)
>
> \[2\] Q. Zhang et. al., How Sparse Can We Prune A Deep Network: A Fundamental Limit Perspective, [https://arxiv.org/abs/2306.05857](https://arxiv.org/abs/2306.05857)
>
> **Calibration-Free Claim.**
> CNNs: We added the pre-REPAIR results in Fig. 17 to provide a complete view of the raw, strictly calibration-free behavior. Our main comparisons use REPAIR to isolate the effect of pruning/folding from variance collapse in BatchNorm layers \- an effect that the original folding paper explicitly recommends correcting via REPAIR or Data-Free REPAIR. Recent work \[3\] reports a similar phenomenon (“signal collapse’’) and likewise applies REPAIR to restore BatchNorm statistics. Following this established practice, we apply REPAIR uniformly to both folded and pruned CNNs using a single unlabeled forward pass \- without labels, optimization, or gradient updates \- so the pipeline remains calibration-free in the standard sense used by calibration-free compression methods.
>
> ViTs: For transformer architectures, we report results immediately after compression in the main paper (Fig. 1), and separately after LN-only tuning (Fig. 2\) and after short full fine-tuning (Fig. 3). Importantly, the post-compression results shown in Fig. 1 require no data and represent the true calibration-free behavior of both methods.
>
> \[3\] D. Saikumar, Signal Collapse in One-Shot Pruning: When Sparse Models Fail to Distinguish Neural Representations, [https://arxiv.org/abs/2502.15790](https://arxiv.org/abs/2502.15790)

---

### Meta-Review · Area_Chair_5YHv · 2025-12-04

**Summary:**

The primary concerns raised by the reviewers were:

1. A lack of experiments on large models
2. Missing comparisons to SoTA
3. Missing runtime analysis
4. Whether the approach is calibration-free

I think the authors provided a strong response and addressed most of these concerns (the SoTA point is questionable however). The original scores of 6462 could feasibly have changed to 8486 (optimistically) or 8464 (conservatively). I recommend acceptance.

**Reviewer Concerns:**

The authors provide experiments with larger LLaMA models, which addresses the concern on model size (and also on domains). With regards to SoTA comparisons, the authors argue that the approaches suggested by the reviewers are fundamentally different in that they are either data-dependent or training-time. I partly buy their reasoning but I would be inclined to compare to the data-dependent approaches in order to show the reader the bigger picture. Runtime analysis is included (in general, the authors have provided a strong response here with experiments). The authors have clarified the “calibration-free” aspect of their approach.

**Reviewer Scores:**

The authors have provided a strong, comprehensive response to Reviewer VBA4, providing a significant amount of analysis. Obviously this is educated guesswork, but I believe the reviewer could well have raised their score to an 8. I think it is unlikely that Reviewer b6Qf would have increased their score without additional pruning experiments. All of Reviewer y5Wu’s points were well answered so there is a chance they would have increased their score. Reviewer kdrS says that all their concerns apart from the point on theory were addressed - their response plus the original review make this read as being higher than a 2 to me.

---

### Decision · Program_Chairs · 2026-01-26

Accept (Poster)